# ^131^I-C19 Iodide Radioisotope and Synthetic I-C19 Compounds as K-Ras4B–PDE6δ Inhibitors: A Novel Approach against Colorectal Cancer—Biological Characterization, Biokinetics and Dosimetry

**DOI:** 10.3390/molecules27175446

**Published:** 2022-08-25

**Authors:** Pedro Cruz-Nova, Blanca Ocampo-García, Dayan Andrea Carrión-Estrada, Paola Briseño-Diaz, Guillermina Ferro-Flores, Nallely Jiménez-Mancilla, José Correa-Basurto, Martiniano Bello, Libia Vega-Loyo, María del Rocío Thompson-Bonilla, Rosaura Hernández-Rivas, Miguel Vargas

**Affiliations:** 1Departamento de Biomedicina Molecular, Centro de Investigación y de Estudios Avanzados del Instituto Politécnico Nacional (CINVESTAV-I.P.N.), Gustavo A. Madero, Mexico City 07360, Mexico; 2Departamento de Materiales Radiactivos, Instituto Nacional de Investigaciones Nucleares, Ocoyoacac 52750, Mexico; 3Laboratorio de Modelado Molecular y Diseño de Fármacos de la Escuela Superior de Medicina, Instituto Politécnico Nacional, Gustavo A. Madero, Mexico City 07360, Mexico; 4Departamento de Toxicología, Centro de Investigación y de Estudios Avanzados del Instituto Politécnico Nacional (CINVESTAV-I.P.N.), Gustavo A. Madero, Mexico City 07360, Mexico; 5Laboratorio de Medicina Genómica, Hospital Regional 1 de Octubre, ISSSTE, Gustavo A. Madero, Mexico City 07360, Mexico

**Keywords:** I-C19, colorectal cancer, K-Ras4B, PDE6δ, pharmacokinetics

## Abstract

In 40–50% of colorectal cancer (CRC) cases, K-Ras gene mutations occur, which induce the expression of the K-Ras4B oncogenic isoform. K-Ras4B is transported by phosphodiesterase-6δ (PDE6δ) to the plasma membrane, where the K-Ras4B–PDE6δ complex dissociates and K-Ras4B, coupled to the plasma membrane, activates signaling pathways that favor cancer aggressiveness. Thus, the inhibition of the K-Ras4B–PDE6δ dissociation using specific small molecules could be a new strategy for the treatment of patients with CRC. This research aimed to perform a preclinical proof-of-concept and a therapeutic potential evaluation of the synthetic I-C19 and ^131^I-C19 compounds as inhibitors of the K-Ras4B–PDE6δ dissociation. Molecular docking and molecular dynamics simulations were performed to estimate the binding affinity and the anchorage sites of I-C19 in K-Ras4B–PDE6δ. K-Ras4B signaling pathways were assessed in HCT116, LoVo and SW620 colorectal cancer cells after I-C19 treatment. Two murine colorectal cancer models were used to evaluate the I-C19 therapeutic effect. The in vivo biokinetic profiles of I-C19 and ^131^I-C19 and the tumor radiation dose were also estimated. The K-Ras4B–PDE6δ stabilizer, ^131^I-C19, was highly selective and demonstrated a cytotoxic effect ten times greater than unlabeled I-C19. I-C19 prevented K-Ras4B activation and decreased its dependent signaling pathways. The in vivo administration of I-C19 (30 mg/kg) greatly reduced tumor growth in colorectal cancer. The biokinetic profile showed renal and hepatobiliary elimination, and the highest radiation absorbed dose was delivered to the tumor (52 Gy/74 MBq). The data support the idea that ^131^I-C19 is a novel K-Ras4B/PDE6δ stabilizer with two functionalities: as a K-Ras4B signaling inhibitor and as a compound with radiotherapeutic activity against colorectal tumors.

## 1. Introduction

Colorectal cancer (CRC) ranks third in terms of cancer incidence and second in terms of mortality, with over 1.8 million new cases and 881,000 deaths estimated in 2018 worldwide [1]. Approximately 40–50% of colon cancers present activating mutations in the K-Ras4B oncogene, which correlates with the progression from benign adenoma to dysplastic colon adenocarcinoma [2]. Activating mutations in K-Ras4B mainly consist of substitutions in G12 and G13 residues, which impair intrinsic GTP hydrolysis mediated by GTP activating protein (GAP). The outcome of these substitutions is the persistence of the GTP-bound state of K-Ras4B and the consequent continuous activation of many different K-Ras4B-dependent downstream pathways [3]. To be stimulated, K-Ras4B requires a translocation from the endoplasmic reticulum to the plasma membrane through the association with the phosphodiesterase 6δ (PDE6δ) protein (K-Ras4B–PDE6δ complex formation), with a subsequent release of activated K-Ras4B into the plasma membrane [4]. It has been demonstrated that the inhibition of the K-Ras4B and PDE6δ association impairs oncogenic K-Ras signaling [5]. However, the stabilization of the K-Ras4B–PDE6δ complex has been recently proposed by our group as a novel strategy to reduce the release of K-Ras4B to the plasma membrane, thus inhibiting K-Ras4B signaling [6,7].

In our previous research, a synthetic small molecule, ((2S)-N-(2,5-dichlorophenyl)-2-[(3,4-dimethoxyphenyl)methylamine]propanamide), known as C19, was found to stabilize the K-Ras4B–PDEδ complex [7]. In this approach, which to our knowledge has not previously been reported by another research group, a synthetic small molecule is used to specifically bind to the interaction site between K-Ras4B and PDE6δ mimicking the effect of a “staple”, thus preventing the K-Ras4B–PDE6δ dissociation. Therefore, the inhibitory effect of C19 causes a decrease in the concentration of K-Ras4B in the plasma membrane and blocks the activation of the mitogen-activated protein kinase (MAPK) signaling pathway. It was also demonstrated that C19 produces a cytotoxic effect in K-Ras4B-dependent LoVo cells by preventing the phosphorylation of ERK and AKT, thus promoting apoptosis and decreasing proliferation [7]. Consequently, and as a new strategy to study its tissue distribution and tumor uptake, chlorine was removed from the C19 structure and replaced with iodine, to be radiolabeled with ^131^I by means of an isotopic exchange. In the broad spectrum of available radionuclides, ^131^I is suitable for in vivo imaging and is an agent with proven radiotherapeutic efficacy due to the energy of the emitted beta particles [8]. The aim of this research was to perform a preclinical evaluation of the (2S)-N-(2,5-diiodophenyl)-2-[(3,4-dimethoxiphenyl)methylamine] propenamide (I-C19 and ^131^I-C19) and establish its uptake, affinity, biokinetics, cytotoxic and radiotoxic effects in colorectal cancer cells and tumors with mutated K-Ras4B.

## 2. Results

### 2.1. In Silico Docking Simulation

In order to evaluate the binding affinity as well as the anchorage sites of the C19 and I-C19 compounds on the wild-type (K-Ras4B^WT^–PDE6δ) and mutant (K-Ras4B^G13D^–PDE6δ) system, docking and molecular dynamics (MD) were performed as previously reported [7]. The complexes formed in molecular docking studies were analyzed by molecular dynamics (MD) simulations to determine the stability. An analysis of the root means squared deviation (RMSD) showed that K-Ras4B^WT^–PDE6δ-C19, K-Ras4B^G13D^–PDE6δ-C19, K-Ras4B^WT^–PDE6δ-I-C19, K-Ras4B^G13D^–PDE6δ-I-C19 complexes reached stable values between 40 and 60 ns, with values oscillating between 1.9 and 3.2 Å (Figure 1A). The evaluation of the radius of gyration (Rg) also showed equilibration at similar simulation times, with values of about 22.5 Å (Figure 1B). Therefore, a further analysis was performed excluding the first 40 ns. To obtain representative protein-ligand conformers, a clustering analysis was performed over the equilibrated simulation time (see methods) considering a 2.5 Å cut-off.

A structural analysis of the most frequent conformations obtained through the clustering analysis showed that C19 within the K-Ras4B^WT^–PDE6δ-C19 complex showed hydrophobic interactions with the residue W90 of PDE6δ and M170 of HVR2 of K-Ras4B^WT^, and eight polar interactions with charged or polar residues of HVR2 of K-Ras4B^WT^ (D173, K177, K180 and S181) and K-Ras4B (Y32, D33, T35 and E37) (Figure 2A). Two hydrogen bonds are formed between C19 and the side chain of D33 and the polar backbone atoms of E37. C19 within the K-Ras4B^G13D^–PDE6δ-C19 showed three hydrophobic contacts with I21, V29 and A59 of K-Ras4B^G13D^, and polar contacts with one reside of PDE6δ (E110) four residues of HVR2 (G174, K177, K178 and S181), and five residues of K-Ras4B (S17, E37, D38, D57 and G60) (Figure 2B). Two hydrogen bonds were formed between C19 and side chains of S17 of K-Ras4B^G13D^ and K177 of HVR2. I-C19 into K-Ras4B–PDE6δ-I-C19 system formed hydrophobic contacts with A59 of K-Ras4B and polar contacts with one residue of PDE6δ (E110), three residues of HVR2 (S181, K177 and K178), and four residues of K-Ras4B (G60, E62, E63 and R68). E63 showed one hydrogen bond with I-C19 through its side chain (Figure 2C). I-C19 forming complex with K-Ras4B^G13D^–PDE6δ makes hydrophobic interactions with A59 of K-Ras4B^G13D^, polar interactions with two residues of PDE6δ (E110 and K57), two residues of HVR2 (K178 and K182), and one residue of K-Ras4B^G13D^ (G60). E110 and K57 also formed hydrogen bonds with I-C19 through their side chains (Figure 2D).

Based on this structural analysis, free binding energy calculations were performed for the K-Ras4B–PDE6δ–C19 [7], K-Ras4B^G13D^–PDE6δ–C19, K-Ras4B–PDE6δ-I-C19 and K-Ras4B^G13D^–PDE6δ-I-C19 complexes, excluding the first 50 ns from the 100 ns MD simulation. Therefore, the ΔG_bind_ values showed that the K-Ras4B/PDE6δ binding was energetically more favorable for the iodinated analog I-C19 (−102.88 kcal/mol) than for C19 (−87.81 kcal/mol) (Table 1). A similar tendency was also observed with C19 (−95.82 kcal/mol) and I-C19 (−149.23 kcal/mol), which formed a complex with K-Ras4BG13D–PDE6δ. These values suggest a higher potential of the I-C19 molecule as a K-Ras4B inhibitor in the wild-type and mutant K-Ras4B–PDE6δ system than that of C19.

The free binding energies and individual energy terms of complexes started from the docked conformations (kcal/mol). The polar (ΔEpolar = ΔEele + ΔGele,sol) and nonpolar (ΔEnon-polar = ΔEvwd + ΔGnpol,sol) contributions are shown (Table 1). All energies were averaged over 500 snapshots at time intervals of 100 ps from the last 50 ns of the MD simulations and are given in kcal/mol (±standard deviation).

The free binding energies and individual energy terms of complexes started from docked conformations (kcal/mol). The polar (ΔEpolar = ΔEele + ΔGele,sol) and nonpolar (ΔEnon-polar = ΔEvwd + ΔGnpol,sol) contributions are shown. All energies are averaged over 400 snapshots at time intervals of 100 ps from the last 40 ns MD simulations and are given in kcal/mol (±standard deviation).

### 2.2. I-C19 Radiolabeling with ^131^I and Evaluation of the Radiochemical Purity

Radioactive labeling of I-C19 with ^131^I (named ^131^I-C19) was performed through an isotopic exchange reaction. The radiochemical purity of ^131^I-C19 was 95% as evaluated by reversed-phase radio-HPLC, and the retention time was 15.3 ± 2 min (Figure 3).

### 2.3. In Vitro Cytotoxicity Studies 

In this study, we investigated the cytotoxic activity and specificity of the analogous compound I-C19 on the colorectal cancer cell lines HCT116, LoVo and SW620, because all of them contain the K-Ras4B mutation, even though the cell line HCT116 is considered K-Ras4B-independent, while the cell lines LoVo and SW620 are K-Ras4B-dependent [9]. The CCD-18Co cell line was also used as a control of healthy human colon cells, which presented an I-C19 IC_50_ of 171.1 µM (data not shown). The HCT116 K-Ras4B-independent cell line showed an IC_50_ of 88.24 µM (Figure 4A), while the K-Ras4B-dependent cell lines had an IC_50_ of 18.39 µM in LoVo cells and 68.28 µM in the SW620 cell line (Figure 4B,C, respectively). The HCT116 and SW620 cancer cell lines did not produce cytotoxic effects in high concentrations (i.e., more than 100 µM; data not shown). This result allowed us to corroborate that the I-C19 compound has a higher affinity than the C19 compound by inhibiting the viability of K-Ras4B-dependent cells at a lower concentration [7]. As shown in the calculation of the binding free energy, I-C19 presented a higher binding energy than C19 (−95.82 vs. −149.23 kcal/mol for C19 and I-C19 in the molecular complex K-Ras4B–PDE6δ, respectively) (Table 1). 

### 2.4. Cell Uptake and Radiation Absorbed Dose

In vitro ^131^I-C19 uptake studies demonstrated a time-dependent cellular accumulation in colorectal cancer cell lines. The total uptake in the cell membrane significantly increased within the first 3 h to 43.6 ± 4.7%, 40.2 ± 6.4% and 45.6 ± 6.7% and decreased at 5 h to 1.63 ± 0.34%, 1.80 ± 0.18% and 25.62 ± 4.71% of the total radioactivity into the HCT116, LoVo and SW620 cells, respectively. Likewise, the internalized activity into the cytoplasm was 2.45 ± 0.21%, 2.90 ± 0.82% and 2.66 ± 0.30 % at 3 h of exposure (Figure 4E) and 1.63 ± 0.05%, 0.30 ± 0.04% and 1.44 ± 0.14% after 5 h of treatment for HCT116, LoVo and SW620 cells, respectively.

Cancer cells were exposed to 1.44 Bq/cell of ^131^I-C19 at a concentration of 8.8, 1.8 and 6.8 µM for HCT116, LoVo and SW620, respectively, and their viability significantly decreased compared to exposure to the I-C19 compound without labeling at the same concentrations after 72 h of treatment (Figure 4D). Therefore, a synergistic cytotoxic effect of chemotherapy (I-C19) and radiotherapy (^131^I-C19) was demonstrated when ^131^I-C19, at absorbed radiation doses of 0.19, 0.29 and 0.43 Gy (total dose from the cytoplasm and cell membrane contributions to the nuclei) significantly affected the viability of HCT116, LoVo and SW620 cells, respectively (Table 2). The inhibition of K-ras4B and its signaling pathways is directly proportional to the radiosensitizing effect on cancer cells. Absorbed radiation doses at 0.2 Gy are enough to induce deterministic effects in human cells, for example, the phenotypic changes observed in blood cells. Therefore, our results demonstrated that a dose of approximately 0.2 Gy is sufficient to effectively potentiate I-C19 cytotoxicity (synergistic effect between I-C19 and I-131 radiation).

### 2.5. DNA Double-Stranded Breaks: γ-H2AX Foci Assay

It is well known that radioactive compounds promote DNA double-stranded breaks (DSBs). DSBs are usually triggered by exogenous chemical agents or ionizing radiation and can lead to genetic instability and apoptosis [10]. DNA damage response (DDR) to ionizing radiation or DNA-damaging chemotherapeutic agents rapidly results in the phosphorylation at serine 139 of histone H2A variant H2AX, named γH2AX. Because γ-H2AX is abundant and correlates well with DSBs, it has been widely used as the most sensitive marker for DSBs for examining the DNA damage produced and the subsequent repair of the DNA lesion [11,12]. Thus, to demonstrate whether the iodinated analogs I-C19 and ^131^I-C19 promote DNA DSBs, immunofluorescence assays were performed to detect the foci of the phosphorylated H2AX histone variant (γ-H2AX). 

The results showed that treatment with I-C19 increased the number of γ-H2AX foci in the cancer cell lines compared with the vehicle (Figure 5A). However, the DSBs foci were significantly higher when HCT116, SW620 and LoVo cells were treated with ^131^I-C19 than with the unlabeled I-C19 (Figure 5A). The radiotoxic effect of ^131^I-C19 was observed at the same level in the K-Ras4B-independent cell line HCT116 as in the LoVo and SW620 K-Ras4B-dependent cells. In K-Ras4B-dependent cell lines (i.e., the SW620 and LoVo cell lines), the γ-H2AX foci increased by 2.3 and 2.1 times with ^131^I-C19 than with the I-C19 compound at the same concentration. However, the HCT116 K-Ras4B-independent cell line showed almost three times more γ-H2AX foci under similar conditions (^131^I-C19 vs. I-C19) (Figure 5B). Consequently, the effect of I-C19 was combined with the effect of the radioactively to promote DNA damage in the K-Ras4B-dependent and independent colorectal cancer cells as well.

### 2.6. GTP-RAS Pull-Down and Human Phospho-Kinase Array

To determine the impact of the I-C19 analog compound on preventing the binding of RAS to GTP as well as on the activation of important kinases in K-Ras4B signaling, a GST-RAS pull-down and human phospho-kinase array were used. The LoVo and SW620 cell lines treated with the IC_50_ of the I-C19 drug concentration showed an 80% decrease in the concentration of RAS-GTP in the LoVo and the SW620 cell lines (Figure 6A; right panel). These results correlate with the results of the binding free energy calculations in which the compound I-C19 presented a ΔG binding of −95.82 kcal/mol when it interacted with the molecular complex K-Ras4B–PDE6δ; however, when the compound I-C19 interacted with the complex K-Ras4Bb^G13D^–PDE6δ (KRAS mutated), the ΔG binding increased (−149 kcal/mol) (Table 1). As expected, the K-Ras4B-independent HCT116 cell line did not show a decrease in active RAS (Figure 6A; left panel). Therefore, a human phospho-kinase array test was carried out to detect whether I-C19 decreases the phosphorylation of kinases involved in K-Ras4B signaling. In the HCT116 cell line, I-C19 treatment reduced the phosphorylation of AKT1/2/3 by 50% (Figure 6B). Nevertheless, the I-C19 compound induced an increase in JNK1/2/3 phosphorylation by 155% which, in turn, promoted the phosphorylation of the CREB and c-Jun transcription factors in HCT116 cells. Additionally, I-C19 treatment reduced the phosphorylation of AKT1/2/3 (−57.7%), ERK 1/2 (−53.4%) and AMPka1 (−58.1%) in the LoVo cell line and reduced the inhibitory phosphorylation of GSK3α/β (S21/S9) involved in WNT signaling (Figure 6C). Moreover, the I-C19 compound reduced the phosphorylation of AKT1/2/3 S473 (−92.6%), AKT1/2/3 T308 (−25%) and AMPka1 (−70%) in the SW620 cell line (Figure 6D). Most notably, the I-C19 treatment reduced the phosphorylation of the WNK1 oncogene (which is overexpressed in colorectal cancer and is mainly involved in the regulation of metabolism) at the levels of 38.6 and 32.6% in the LoVo and SW620 cell lines, respectively [13]. Indeed, I-C19 treatment reduced the concentration of GTP-RAS and, therefore, decreased the phosphorylation of kinases involved in K-Ras4B signaling in K-Ras4B-dependent cell lines. 

### 2.7. I-C19 Compound Suppressed Tumor Growth In Vivo

To determine whether the I-C19 compound induced a decrease in neoplastic cells in vivo, the in situ model of colorectal cancer with AOM/DSS was used. At week 50 after injection of 10 mg/kg of AOM and two cycles of DSS, the histological analysis showed the development of malignant neoplastic lesions of glandular epithelial lineage in mice colorectal tissue. Aberrant glands formed these lesions, and the cells that formed them presented atypical nuclei with evident nucleoli and loss of mucus production. The percentage of mice colorectal tissue affected with these lesions was considered as the percentage of colon adenocarcinoma in mice.

After 12 days of treatment with the vehicle (i.e., PBS 1X with 10%DMSO), the percentage of colon adenocarcinoma in mice was 7–8%, and mice treated with C19 (30 mg/kg) showed a percentage of colon adenocarcinoma of 5–6%. However, I-C19 (30 mg/kg) treatment significantly reduced the rate of colon adenocarcinoma (3–4%) (Figure 7A). In the AOM/DSS mice model treated with I-C19 for 12 days, no changes were observed in body weight (Figure 7B). To evaluate the effect of I-C19 on the degree of tumor malignancy and the proliferation of colon cancer cells, the immunohistochemistry assay using carcinoembryonic antigen (CEA) and intrinsic proliferation marker Ki67 was conducted. I-C19 treatment significantly reduced the CEA marker (7.5% of neoplastic tissue) compared to treatment with C19 (12.5% of neoplastic tissue) and the vehicle (65% of neoplastic tissue) (Figure 7C; upper panel). In addition, I-C19 reduced the proliferation of neoplastic cells in the colon of mice by 20% as measured with the Ki67 marker (Figure 7C; bottom panel). 

To further investigate this point, the LoVo CRC xenograft model was developed in nude mice. Mice were subsequently treated with 30 mg/kg I-C19 and C19 for 12 days, and the tumors were measured with a caliper. As can be seen in Figure 8, from days 10 to 12, I-C19 (~160 mm^3^) and C19 (~200 mm^3^) treatments significantly inhibited tumor growth (Figure 8A), and I-C19 effectively reduced the tumor size by six-fold relative to the control that reached ~1000 mm^3^ (Figure 8B). Moreover, the I-C19 compound did not reduce body weight during the 12 days of treatment (Figure 8C). In the immunohistochemical analysis, CEA-positive cells increased in tumors treated with the vehicle, and I-C19 treatment significantly reduced the expression of the CEA marker (Figure 8D,E; upper panel). The treatment of tumor-bearing mice with I-C19 suppressed the proliferation of Ki67-positive cells (Figure 8D,E; middle panel). Based on our previous study, which found that C19 triggers apoptosis in vivo, we hypothesized that the I-C19 compound might exert antitumor activity through induction of apoptosis. It is worth noting that we detected an increase in cleaved caspase-3 in tumors treated with I-C19 (Figure 8D,E; bottom panel).

### 2.8. Biodistribution

Biodistribution studies showed both renal and hepatobiliary excretion of ^131^I-C19. The radiopharmaceutical study showed good specific tumor uptake of LoVo cells in athymic mice. In all other organs, the average percentage of injected radioactivity (%IA) of ^131^I-C19 decreased in parallel after the administration of 5 MBq (135 µCi/mice) of ^131^I-C19 (Figure 9A,B). The ^131^I-C19 radiopharmaceutical was cleared from the blood according to the $A_{h\;}\left(t\right)=89.8e^{-1.524t}+10.2e^{-0.033t}$ fitted model, which showed that 89.8% of the activity was rapidly cleared from the blood within 0.45 h (ln2/1.524 = 0.45 h) and the remaining 10.2% was cleared within 21 h (ln2/0.033 = 21 h). I-C19 was also cleared rapidly from the blood as shown in the fitted biokinetic model: $q_{h\;}\left(t\right)=89.8e^{-1.520t}+0.260e^{-0.029t}$ in which 89.8% of the activity was rapidly cleared from the blood within 0.46 h (ln2/1.520 = 0.46 h) and the remaining 10.2% was cleared within 24 h (ln2/0.029 = 24 h). The lung and liver were the organs that received the highest absorbed doses of radioactivity: 0.40 and 0.34 Gy, respectively; the colon received 0.14 Gy and the kidney received 0.12 Gy (Table 3). Because the reported maximum tolerated dose for the lungs and liver is 25–40 Gy, severe radiotoxicity was not expected after ^131^I-C19 intraperitoneal administration. The radiation dose delivered to the colon was lower than that of the spleen (0.19 Gy) and pancreas (0.18 Gy). The highest absorbed dose was observed in the tumor (3.55 Gy/5 MBq) (Table 3), which means that the administration of 74 MBq (2 mCi) would produce ablative radiation doses (52 Gy/74 MBq).

### 2.9. Genotoxicity Test (Micronucleus Assay)

To determine if the treatment could represent a genotoxic risk (i.e., increased damage to DNA), the structural chromosomal damage (i.e., clastogenicity) or lagging chromosomes in the erythrocytes of bone marrow (i.e., aneuploidogenesis) and the frequency of micronucleated polychromatic erythrocytes (MNPCEs) were evaluated [14]. A pronounced increase in the frequency of MNPCEs was observed with the treatment of 5-fluorouracil (5-Fu) (27 ± 4) compared to the vehicle (7 ± 2) (Figure 10A,B; left panel). The mean values of the data obtained from 24 h of treatment showed an increase but not a significant number of MNPCEs in the presence of 30 mg/kg of C19 (12 ± 1) and I-C19 (11 ± 2) (Figure 10A,B; right panel). These data suggest that exposure to C19 or I-C19 in vivo does not increase damage to DNA or the induction of aberrant chromosome distribution into the daughter cells.

## 3. Discussion

In a previous work, the therapeutic effect of the C19 molecule against colorectal cancer was demonstrated [7]. That research suggested that C19 interacts with the K-Ras4B–PDE6δ protein complex by increasing the binding energy of the K-Ras4B–PDE6δ complex. The ΔG (binding) of the K-Ras4B-C19–PDE6δ system was −87.81 kcal/mol, blocking its dissociation and functions [7]. More significantly, this compound showed an increase in the binding energy and stabilization of the mutant K-Ras4B^G13D^–PDE6δ protein complex with a ΔG (binding) of −95.82 kcal/mol, suggesting that this mechanism inhibits the activity of the mutated KRas4B protein. In this work, we demonstrated that the exchange of the chlorine for iodine in the C19 molecule improved the ΔG (binding) of the K-Ras4B-I-C19–PDE6δ system, which increased from −95.82 kcal/mol to −149.23 kcal/mol in the mutant K-Ras4B^G13D^-I-C19–PDE6δ system (Table 1). 

This result suggests that I-C19 strongly binds to the mutant K-Ras4B–PDE6δ complex, preventing its dissociation and activity. This highly selective inhibitor of PDE6δ presents greater affinity and produces less nonspecific cytotoxicity compared with other PDE6δ inhibitors, such as deltarasin [15]. In this work, I-C19 showed high specific cytotoxicity; the IC_50_ value of the healthy CCD-18Co cell line after treatment with C19 was 60.8 µM [7], and a significantly increased IC_50_ was measured (171.1 µM) with the I-C19 treatment (data not shown). 

The lowest IC_50_ value was obtained in the LoVo cell line (18.39 µM) and was one order of magnitude lower than the IC_50_ obtained in healthy human colon cells (171.1 µM). The IC_50_ in the SW620 cell line was 2.5 times lower than the IC_50_ value obtained in the control cell line (68.28 vs. 171.1 µM). Compared to the IC_50_ value in the K-Ras4B-independent HCT116 cell line, the obtained response in LoVo cells was five times higher, and the effect produced in the SW620 cell line was only 1.2 times higher than in HCT116.

The main drawback of cancer therapy (immunotherapy or small molecules) is the poor penetration of drugs into solid tumors, including colorectal cancer tumors. To circumvent this problem, the conjugation of drugs to α- or β^-^-emitting radionuclides can significantly improve their therapeutic effects [16]. In this work, I-C19 radiolabeled with ^131^I exhibited a higher cytotoxicity at a concentration ten times lower than the IC_50_ of unlabeled I-C19 in three different colorectal cancer cell lines (Figure 4D). Consequently, in Na[^125^I] uptake trials, it was observed that at 1 h posttreatment, three murine cell lines of CMT93 (colorectal cancer), EMT6 (breast cancer) and CMT64 (lung cancer) displayed significantly increased iodine-conjugate accumulation [17]. The radiolabeled compound ^131^I-C19 showed an initial uptake of 3.28–3.6%, while after 3 h of treatment, colorectal cancer cell lines displayed a significantly increased uptake (Figure 4E). The high uptake of ^131^I-C19 at three hours of treatment suggests that the beta emission of [^131^I] (β-, Emax = 606 keV) could decrease the viability of cancer cells through DNA double-stranded breaks (DSBs). Indeed, ^131^I-C19 promoted DNA DSBs, which was evidenced by the significant accumulation of γH2AX foci in the colorectal cancer cell lines compared to the I-C19-treated and untreated cells (Figure 5A,B). DNA DSBs are highly dangerous lesions with severe consequences for cell survival. DSBs in chromatin promptly initiate the phosphorylation of the histone H2A variant H2AX at serine 139 to generate γH2AX [18]. LoVo cells were more radioresistant than the SW620 KRAS-dependent cells [19], which may explain why the DSB concentrations were two times higher in LoVo than in SW620 cells.

Other studies demonstrate that a β-emitter can significantly decrease the proliferative capacity through the induction of DNA DSBs [20]. The high level of DSBs in HCT116 (six times higher than the control) due to the fact of exposure to ^131^I beta radiation was expected, because the uptake did not show statistically significant differences among all evaluated cells. However, the radiation dose absorbed in LoVo cells was higher than that of the HCT116 and SW620 cells (i.e., 0.19, 0.29 and 0.43 Gy/Bq, respectively).

Targeting K-Ras4B by disrupting its cellular localization or posttranslational modifications has successfully harnessed K-Ras4B for clinical therapeutics [21]. The main objective of the I-C19 compound is to inhibit K-Ras4B activation, which occurs as shown in Figure 6A. The phosphorylation statuses of proteins involved in the downstream signaling pathway and other essential pathways in cancer, such as angiogenesis, survival and anti-apoptosis, proliferation, metabolism, chemoresistance and DNA repair, are important [7]. For example, Yang et al. (2021) describes the signaling pathway by which mutated KRAS can cause cells to acquire radioresistance. They demonstrated that KRAS mutations result in a more significant response to DNA damage and the positive regulation of 53BP1 (a protein associated to dsDNA repair) with increased associated nonhomologous end-junction repair (NHEJ). Moreover, KRAS mutations lead to the activation of NRF2 antioxidant signaling to increase the transcription of the 53BP1 gene [22]. Additionally, AKT-dependent WNK1 phosphorylation and enhanced WNK functions (such as increased phospholipase C-β signaling, angiogenesis and cell migration) have been reported. Here, after I-C19 treatment in K-Ras4B-dependent cell lines, AKT and WNK1 decreased phosphorylation (Figure 6C,D, respectively). Moreover, one of the pathways that is upregulated in colorectal cancer is the mTOR pathway. It has been shown that RHEB (a Ras homolog enriched in the brain) silencing leads to a decrease in mTOR, P70S6K and 4EBP1 phosphorylation and the expression of RHEB, Ki67, mTOR, P70S6K, BCL-2 and PCNA as well as cell proliferation and differentiation [23]. Treatment with I-C19 showed a significant decrease in RAS-dependent pathway activity and, therefore, the phosphorylation of AKT1/2/3, PRAS40, P70S6K and ERK1/2 (essential kinases in anti-apoptosis and proliferation pathways) was diminished. Furthermore, after I-C19 treatment, the phosphorylation of AMP-activated protein kinases, AMPKa1 and AMPKa2, decreased only in the K-Ras4B-dependent cell line (Figure 6D). Our results showed that I-C19 significantly inhibits the activation of K-ras4b and its signaling pathways in K-ras4b-dependent colorectal cancer cells (Figure 6A,C,D). However, there is a possibility that I-C19 is inhibiting the binding of K-ras4B to GTP by a pathway other than stabilization of the K-ras4b-PDE6δ complex. Therefore, as future work, it is necessary to demonstrate the activity of the I-C19 compound at the molecular level, for example, by means of isothermal titration calorimetry (ITC) assays with recombinant proteins.

These findings suggest that treatment with I-C19 in combination with radiotherapy can be a powerful tool for treating certain types of K-Ras4B-dependent cancers. Furthermore, we analyzed the therapeutic potential of I-C19 in colorectal cancer in vivo. We used azoxymethane, a rodent colon-specific carcinogen that induces DNA damage and causes proto-oncogene K-Ras4B point mutations and subsequent tumor formation if DNA damage is not repaired or removed [24]. We were able to observe that a lower percentage of I-C19-treated mice developed colorectal adenocarcinomas compared to mice treated with the vehicle or C19 (Figure 7A), suggesting that I-C19 decreases the activation of K-Ras4B in vitro and in vivo and that the radiation delivery system through C19 to colon cancer cells is efficient, as it also reduces CEA (Figure 7C; upper panel) and Ki67 staining (Figure 7C; lower panel), which allowed us to confirm that this treatment decreased the malignancy and proliferative capacity of colorectal cancer with mutated K-Ras4B. This experimental evidence agrees with the theoretical results that suggested that I-C19 strongly binds to the mutant K-Ras4B–PDE6δ complex, avoiding its dissociation and activity. These findings are consistent with those reported by Shen and collaborators in 2017 [25], who identified miR-30a as a synthetic lethal agent in K-Ras4B-mutant CRC cells. miR-30a directly targeted ME1 and K-Ras4B and inhibited anchorage-independent growth and in vivo tumorigenesis by K-Ras4B-mutant CRC cells. These treatments showed reduced proliferation and increased apoptosis as indicated by the decrease in Ki67 and increase in cleavage caspase-3 staining [26]. This finding correlates with our results. After I-C19 treatment, we were able to observe a decrease in tumor growth in the LoVo xenograft model. I-C19 decreased malignancy and proliferation and increased apoptosis as demonstrated by the decrease in CEA and Ki67 and the increase in cleavage caspase-3 staining (Figure 8D,E).

To confirm that I-C19 was taken up by the tumor cells and to evaluate the route of excretion, we performed biodistribution studies with ^131^I-C19. We observed higher uptake rates of ^131^I-C19 in the tumor and an accumulation in the liver and kidneys, suggesting that the compound is metabolized by the liver and eliminated through the urine. Similarly, biodistribution studies after injection of Na[^125^I] in mice bearing CMT93 (mice colorectal cancer) allograft tumors resulted in 9.1% ID/g in tumor tissue at 30 min post-injection [17]. In this study, after 1 h, the ^131^I-C19 activity concentration in the tumor tissue was 4.4% ID/g. These findings correlate with the reduction in tumor growth in mice treated with I-C19. The highest radiation dose was delivered to the tumor (3.55 Gy/5 MBq).

Finally, the increase in incidences of micronuclei formation is a useful biomarker of genotoxicity, either through the clastogenic mechanism (i.e., direct DNA damage) or through secondary interaction with DNA replication (i.e., indirect aneugenic mechanism) [14]. Cyclophosphamide (CP) is a potent antitumor agent used globally against many forms of human cancers. When mice were treated with 30 mg/kg of CP, the induction of micronuclei was strong [26]. Similarly, when we treated mice with 15 mg/kg of 5-Fu (a widely used chemotherapeutic), a significant increase in MNPCEs occurred (Figure 10). Although this is expected from classical chemotherapeutic drugs, it is a secondary deleterious effect that reduces the quality of life of many patients and can lead to the production of other de novo tumors in patients over time. In contrast, the in vivo induction of micronuclei in PCEs treated with C19 or I-C19 was similar to the levels of micronuclei induction with the vehicle (Figure 10), indicating that the C19 or the I-C19 compounds did not represent an elevated genotoxic risk as do the classical chemotherapeutic agents.

## 4. Materials and Methods

### 4.1. Compounds and In Silico Molecular Analysis

The C19 compound [7] was modified by halogen exchange to synthesize I-C19 ((2S)-*N*-(2,5-diiodophenyl)-2-[(3,4-dimetoxyphenyl)-methylamine] propanamide), which was acquired from Enamine (Kyiv, Ukraine). In silico docking simulation, free binding energies calculation and molecular dynamics simulations for I-C19 and C19 that formed complexes with K-Ras4B/PDE6δ and K-Ras4B^G13D^/PDE6δ were performed as previously reported [6,7,27]. The coupling calculations were performed using 2.5 × 10^5^ positions for the I-C19 compound on a specific target. Molecular docking was performed using the “matcher” function to produce the initial poses. The results from the London dG score were refined using energy minimization (MMFF94x force field) and rescored using Affinity dG scoring. The AMBER 16 package and the ff14SB force field were used to perform the protein–ligand molecular dynamics simulations. The free binding energies were carried out using the MMGBSA (Amber16 suite) [28], following the procedure detailed in our previous research [6,7,29]. The trajectories were analyzed using cpptraj tool in Amber16 to determine the time-dependence of the RMSD, radius of gyration (RG), and clustering analysis.

### 4.2. I-C19 Radiolabeling with ^131^I

The electrophilic radioiodination of the I-C19 compound was performed using chloramine T as an oxidizing agent. Ten batches of ^131^I-C19 were prepared using solutions containing 400 µg (700 µmol) of I-C19 to which 12 MBq (50 µL) of [^131^I]-NaI and 50 µL of chloramine T (1 mg/mL) was added. After 2 min of vigorous agitation, 20 µL of a 15 µM Na_2_S_2_O_5_ solution was added to interrupt the reaction. The final product, ^131^I-C19, was purified by solid-phase extraction on a Sep-Pak C_18_.

### 4.3. Radiochemical Purity Evaluation

The radiochemical purity of ^131^I-C19 was evaluated on a reversed-phase HPLC system (Waters Corporation, Milford, Massachusetts, USA) with a UV photodiode array detector and a radioactivity detector. To separate the components of the mixture (i.e., unlabeled I-C19 and ^131^I-C19), a µBondapak C18 column (3.9 mm × 300 mm, 5 µm) was used. A gradient at 1 mL/min of TFA:water (0.1%, *v*/*v*, phase A) and TFA:acetonitrile (0.1%, *v*/*v*, phase B) was used. The gradient started at 100% of A and was maintained for 3 min before being changed to 50% of A (3–23 min) and, finally, 100% of A was maintained for 27–30 min.

### 4.4. Human Colorectal Cancer Cell Lines

The human CRC cell lines, LoVo (ATCC^®^ CCL-229^™^), SW620 (ATCC^®^ CCL-227^™^) and HCT116 (ATCC^®^ CCL-247™), were acquired from ATCC and cultured in F12K medium (Kaighn’s Modification of Ham’s F-12 Medium), DMEM-F12 medium and RPMI-1640 medium, (ATCC, Manassas, VA, USA), respectively. The human colorectal fibroblast CCD-18Co line was grown in an EMEM medium (ATCC, Manassas, VA, USA). The media were added with 10% fetal bovine serum and a 1% penicillin/streptomycin solution. Cells were incubated at 37 °C in a humidified atmosphere of 5% CO_2_.

### 4.5. Cell Uptake

The HCT116, LoVo and SW620 cells were seeded in 48-well tissue culture plates at a density of 10^5^ cells/well. After 24 h, the cells were exposed to 1.44 Bq/cell of ^131^I-C19 (200 μL) at 37 °C for 1, 3 and 5 h. After treatment, two washes (1 mL of ice-cold PBS) were performed. For the calculation of the cell uptake, the cells in each well were treated (at room temperature for 5 min) with 500 μL of glycine buffer (50 mM, pH 2.8). Then, this fraction was removed, and the radioactivity was measured in a NaI (Tl) detector (BCL, San Rafael CA, USA). To evaluate the ^131^I-C19 internalization, the cells were incubated with 1 mL of NaOH 0.3 M (at room temperature for 10 min). Once more, the radioactivity was measured in a NaI (Tl) detector (BCL, San Rafael CA, USA). A 100% standard solution (200 μL) of ^131^I-C19 was also measured (n = 5). The uptake (%) (plasmatic membrane) and internalization (%) (cytoplasm) were used to obtain the radiation absorbed dose.

### 4.6. I-C19 Cell viability

The CCD-18Co, SW620, LoVo and HCT116 cell lines were plated in 96-well plates at a density of 10^4^ cells/well for 24 h to allow for adherence. Then, the cells were treated with the I-C19 compound for 72 h. Cell viability was evaluated by the XTT (2,3-bis[2-methoxy-4-nitro-5-sulfophenyl]-2H-tetrazolium-5-carboxyanilide) assay kit (Roche, Germany), based on to the manufacturer’s instructions. The required I-C19 concentration, which produced fifty percent cell death at 72 h, was calculated by fitting a model with Prism software (GraphPad Software, San Diego, CA, USA).

### 4.7. 131 I-C19 Cell Viability

For the assessment of dual therapy, beta radiation emission from ^131^I-labeled C19 was also evaluated. The mitochondrial dehydrogenase activity in living HCT116, LoVo and SW620 cells was considered to measure the contribution of ^131^I beta radiation to cell death, and the cell viability analysis was carried out after exposure to concentrations ten times lower than the IC_50_ obtained for each cancer cell line (8.8, 1.8 and 6.8 µM, respectively). Cell viability was evaluated by the XTT (2,3-bis[2-methoxy-4-nitro-5-sulfophenyl]-2H-tetrazolium-5-carboxyanilide) assay kit (Roche, Germany), based on to the manufacturer’s instructions. Briefly, colorectal cancer cell lines at a density of 10^4^ cells/well were seeded in 96-well microtiter plates. The viability after treatment was measured 72 h (incubation in 5% CO_2,_ 85% humidity and 37 °C). The absorbance in each well was recorded in a microplate reader (Epoch; BioTek Instruments; Winooski, VT, USA) at 450 nm. The cells without treatment were considered the control group (100% viability).

### 4.8. Radiation Absorbed Dose to the Colon Cancer Cell Nucleus

For estimation of the radiation absorbed doses to the HCT116, LoVo and SW620 cells’ nuclei, Equation (1) was used:(1)$$D_{N\leftarrow source}=(N_{CS}\times DF_{N\leftarrow CS})+\left(N_{Cy}\times DF_{N\leftarrow Cy}\right)$$
where $D_{N\leftarrow source}$ denotes the mean absorbed dose to the nucleus from source regions (i.e., cell surface—CS; cytoplasm—Cy); $N_{CS}$ and $N_{Cy}$ are the total nuclear transformations occurring in the membrane and cytoplasm, respectively, calculated from the mathematical integration (from *t* = 0 to *t* = ∞) of the activity estimated in the internalization and cell uptake experiments; $DF_{N\leftarrow CS}$ and $DF_{N\leftarrow Cy}$ represent the dose factors specific for [^131^I] from the cell surface (plasmatic membrane) and cytoplasm regions to the nucleus configuration. The dose factor geometries were obtained from S values [30] (HCT116 cell radius = 10 μm, nucleus radius = 5 μm; LoVo and SW620 cell radius = 10 μm, nucleus radius = 6 μm).

### 4.9. DNA Double-Stranded Breaks Assay

For the evaluation of DNA double-stranded breaks (DSBs), the phosphorylation of the Ser-139 gamma histone variant H2AX (γH2AX) was used as a biomarker. Cells, diluted to a suitable concentration, were grown on a glass slide after exposure to I-C19 or ^131^I-labeled C19 (1.8, 6.8 and 8.8 µM for the cell lines LoVo, SW620 and HCT116, respectively) for 3 h. The anti-phospho-histone H2A.X (ser139) (1:200; CST, Boston, MA, USA) antibody was used as the primary antibody. The Alexa-488-conjugated anti-rabbit IgG was used for the visualization of γH2AX. Afterwards, the slides were stained with DAPI, and the foci were observed with a Leica confocal microscope. To quantify the foci, clear and easily distinguishable dots were counted. At least 30 cells were imaged per slide, and the average number of foci per cell was calculated.

### 4.10. RAS Activation Level Assay

For the evaluation of the RAS activation level after treatment with the I-C19 compound, the RAS-GTP pull-down assay was carried out via Western blotting using the RAS Activation Assay Biochem Kit (BK008; Cystoskeleton, Inc., Denver, CO, USA), according to the standard procedure. Briefly, HCT116, LoVo and SW620 cells were grown in to obtain 3 × 10^6^ cells and then lysed in ice-cold lysis buffer (400 µL) supplemented with cOmplete^™^ EDTA-free Ultra Protease Inhibitor Cocktail and PhosSTOP^™^ (Sigma-Aldrich, St. Louis, MO, USA). Lysates were centrifuged, and the protein (300 µg) was collected. Lysates were incubated by end-over-end rotation with 100 µg of Raf-RBD-conjugated beads for 1 h. The supernatant was removed, and the beads were washed and boiled in Laemmli sample buffer, followed by Western blot analysis using the pan-RAS antibody.

### 4.11. Phospho-Kinase Downstream Activation of K-Ras4B Signaling

To evaluate the status of phospho-kinase downstream activation of K-Ras4B signaling after I-C19 treatment, cells were rinsed with cold PBS and immediately lysed in a buffer supplemented with Complete^™^ EDTA-free Ultra Protease Inhibitor Cocktail (Sigma-Aldrich) and PhosSTOP^™^ (Sigma-Aldrich) at 4 °C for 30 min. After centrifugation (14,000 *g*, 5 min), supernatants were transferred to a clean tube, and protein concentrations were quantified using the Precision Red Advanced Protein Assay (Cytoskeleton, Inc., ADV02-A). The lysates were diluted and analyzed using the Human Phospho-Kinase Antibody Array (Proteome Profiler, R&D Systems, Minneapolis, MN, USA). Nitrocellulose membranes were scanned using ChemiDoc^TM^ Imaging Systems (BIO-RAD Laboratories, Inc., Hercules, CA, USA).

### 4.12. Experimental Animals

All mice were supplied by the Laboratory Animals Production and Experimentation Unit of CINVESTAV-Zacatenco. The Institutional Animal Care and Use Committee of CINVESTAV (Protocol No. 0238-17) previously approved the animal trials. The mice were housed in the CINVESTAV animal facility under standard conditions (i.e., temperature 21° ± 2, humidity of 50% ± 10% and a 12:12 light:dark cycle). At the end of all treatments, the mice were euthanized by CO_2_ asphyxiation.

### 4.13. Colorectal Cancer Mouse Model (Azoxymethane/Dextran Sulfate Sodium)

Mice were treated as previously described [24]. Six-week-old C57BL/6 male mice, with a weight of approximately 20 g, were intraperitoneally injected with a single dose of azoxymethane (AOM) (10 mg/kg). One week later, the mice were given a course of 1.5% dextran sulfate sodium (DSS) in sterile drinking water for 7 days, followed by pure drinking water for two weeks for a total of three courses. Mice were then randomly divided into three equal groups: vehicle (10% DMSO in PBS); I-C19 (30 mg/kg); C19 (30 mg/kg). For the control group, the same procedure was performed with normal intraperitoneal saline and pure drinking water instead of the AOM/DSS treatment.

### 4.14. Xenograft Model: Subcutaneous Injection and Preclinical Drug Testing

Six-week-old nu/nu male mice, with weights of approximately 20 g, were used. A suspension of LoVo cells (5 × 10^6^/200 µL PBS) was subcutaneously injected into the right flanks of nude mice. When tumors reached an average volume of 150 mm^3^, the mice were randomly assigned to two groups. The first group was administered with a vehicle (i.e., 10% DMSO in PBS) and the second was administered with I-C19 (i.e., 30 mg/kg). Treatments were administered intraperitoneally, in a final volume of 300 µL, once daily, for 12 days. Tumor volumes were calculated using the formula mm^3^ = (½ d)(D^2^), with d and D being the shortest and longest diameters, respectively.

### 4.15. Immunohistochemistry

Paraffin sections of colon tumor samples were deparaffinized in xylene and rehydrated in a series of graded alcohols. Antigens were retrieved in 0.01 M sodium citrate buffer and 0.1M EDTA-Tris buffer (Novocastra^™^ Epitope Retrieval Solutions, cat. RE7113; Leica Biosystems). Samples were incubated in 0.9% H_2_O_2_ for 5 min, followed by 1 h of blocking in 1% BSA in PBS. Slides were incubated for 1 h at room temperature with anti-Ki67 (1:100, Clone SP6 CRM325, BIOCARE Medical), anti-cleaved caspase 3 (1:100, Asp175 9661, CST) and anti-carcinoembryonic antigen (1:50; Clone COL-1, BIOCARE Medical) and washed and incubated in MACH1 Universal HRP-polymer (Cat. MRH538G; BIOCARE Medical) for 1 h at room temperature. Then, the samples were developed with the Betazoid DAB Chromogen Kit (Cat. BDB2004H; BIOCARE Medical), counter-stained with hematoxylin and mounted with synthetic resin solution (Entallan.107960; Merck).

### 4.16. Biodistribution and Biokinetic Models

^131^I-C19 was intraperitoneally injected (200 µL, 5 MBq (135 µCi/mice)) into nu/nu mice. The mice (n = 3) were euthanized at 0.5, 1, 3, 5, 24, 48, 92 and 144 h post-injection. Whole tumor, lung, liver, pancreas, spleen, kidney, large intestine (colon) and blood were rinsed and placed into pre-weighed plastic test tubes. The activity was determined in a NaI(Tl) detector (Canberra) along with three 0.5 mL aliquots of the diluted standard, representing 100% of the injected activity and used for radioactive decay correction. Mean activity values in each organ were correlated with those of the standard (representing the initial injected activity) to obtain the percentage of injected dose (% ID) at different times. The % ID values of each organ or tumor were input into the Olinda/EXM 2.0 software and adjusted to exponential functions (first-order kinetics) to obtain biokinetic models corrected by radioactive decay ($q_{h}\left(t\right)$) (Equation (2)), which corresponded to the I-C19 compound. Likewise, the $A_{h}\left(t\right)$ functions, which corresponded to the ^131^I-C19 biokinetic models, were obtained considering $\lambda_{eff}$ as $=\lambda_{B}+\lambda_{R}$, where the biological constant is represented by *λ_B_* and the ^131^I−radioactive decay constant is *λ_R_* (Equation (3)).
(2)$$q_{h}\left(t\right)=A_{h}\left(t\right)\times e^{\lambda_{R}}\;=\;Be^{(-\lambda_{B_{b}})t}+Ce^{(-\lambda_{B_{c}})t}+De^{(-\lambda_{B_{d}})t}\;$$
(3)$$A_{h}\left(t\right)=Be^{-\left(\lambda_{B_{b}}+\lambda_{R}\right)t}+Ce^{-\left(\lambda_{B_{c}}+\lambda_{R}\right)t}+De^{-\left(\lambda_{B_{d}}+\lambda_{R}\right)t}\;$$

### 4.17. Radiation Absorbed Dose Estimation

The mathematical integration of $A_{h}\left(t\right)$ functions yielded the total nuclear transformations (MBq^.^s) that occurred in each source organ (Equation (4)).
(4)$$N_{source}=\int_{0}^{\infty }A_{h}\left(t\right)dt$$

The absorbed doses delivered by ^131^I-C19 to selected organs were estimated based on the following Equation (5):(5)$$D_{target\leftarrow source}=N_{source}\times DF_{target\leftarrow source}$$
where $D_{target\leftarrow source}$ is the mean absorbed dose to the target organ from a source organ; $DF_{target\leftarrow source}$ is a dose factor that considers the fraction of absorbed energy for each nuclear emission of iodine-131 in the different geometries and chemical compositions of the organs. Iodine-131 DF values (mGy/MBq·s) for a 25 g mouse model were obtained from the Olinda code, version 2.2. For the tumor dose calculation, the sphere model was used considering an average tumor mass of 0.1 g (Olinda 1.2 software).

### 4.18. Genotoxicity Assay

To evaluate the genotoxic effect of I-C19, a micronucleus assay with bone marrow cells was carried out according to the method previously described [31]. The test compounds were intraperitoneally administered once, as a solution, at a concentration of 15 mg/kg of 5-fluorouracil (5-Fu) and 30 mg/kg of C19 or I-C19. Bone marrow cells were obtained 24 h after treatment and stained with Giemsa-Wright (Diff-Quick; Harleco; Gibbstown, NJ, USA). Two-thousand polychromatic erythrocytes were counted per animal using an optical microscope at 100× magnification to determine the number of micronucleated polychromatic erythrocytes.

### 4.19. Statistical Analysis

Statistical comparisons were performed with one-way analysis of variance (ANOVA) followed by Dunnett’s multiple comparisons test, using GraphPad Prism 5.0 software. The data are shown as the mean ± SEM. A *p*-value < 0.0001 represent as (****) in the graphs and *p*-value < 0.01 represent as (*) in the graphs, was considered statistically significant.

## 5. Conclusions

Based on in vitro and in vivo evaluations, it is conceivable to design a formulation and continue preclinically and clinically assessing I-C19 and ^131^I-C19 as potential molecules for the treatment of K-Ras4B-dependent or K-Ras4B-independent colon cancers.

## Figures and Tables

**Figure 1 molecules-27-05446-f001:**
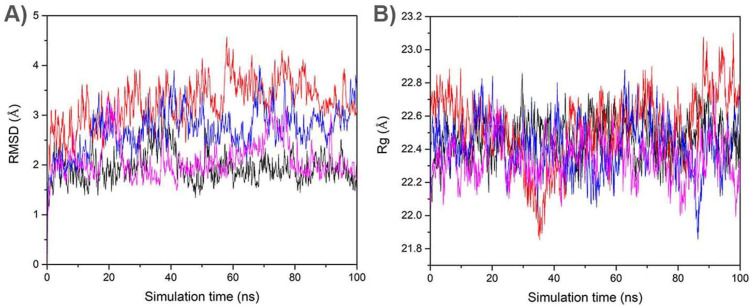
(**A**) RMSD and (**B**) Rg values of the complexes between KRas4B^WT^–PDE6δ or KRas4B^G13D^–PDE6δ with C19 or I-C19. RMSD (left) values of KRas4B^WT^–PDE6δ-C19 (black line) KRas4B^G13D^–PDE6δ-C19 (red line), KRas4B^WT^–PDE6δ-I-C19 (blue line), and KRas4B^G13D^–PDE6δ-I-C19 (magenta line). Rg (right) values of KRas4B^WT^–PDE6δ-C19 (black line), KRas4B^G13D^–PDE6δ-C19 (red line), KRas4B^WT^–PDE6δ-I-C19 (blue line) and KRas4B^G13D^–PDE6δ-I-C19 (magenta line).

**Figure 2 molecules-27-05446-f002:**
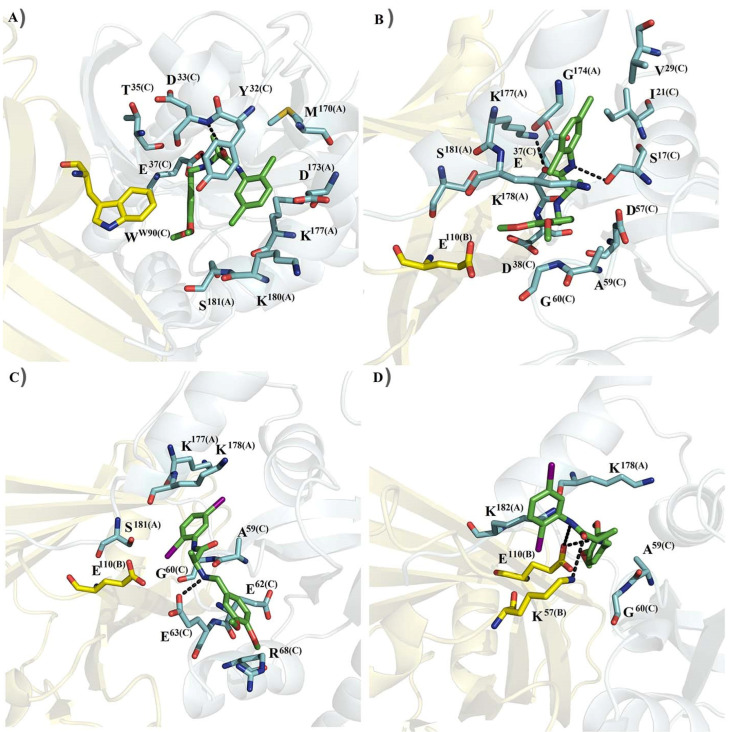
Most frequent conformations of C19 and I-C19 within the K-Ras4B–PDE6δ (cyan–yellow) complex during the MD simulations: (**A**) K-Ras4B–PDE6δ–C19; (**B**) K-Ras4B^G13D^–PDE6δ-C19; (**C**) K-Ras4B–PDE6δ–I-C19; (**D**) K-Ras4B^G13D^–PDE6δ–I-C19. Labels A, B and C denote residues from HVR2, PDEδ and K-Ras4B, respectively. The dotted lines represent hydrogen bonds.

**Figure 3 molecules-27-05446-f003:**
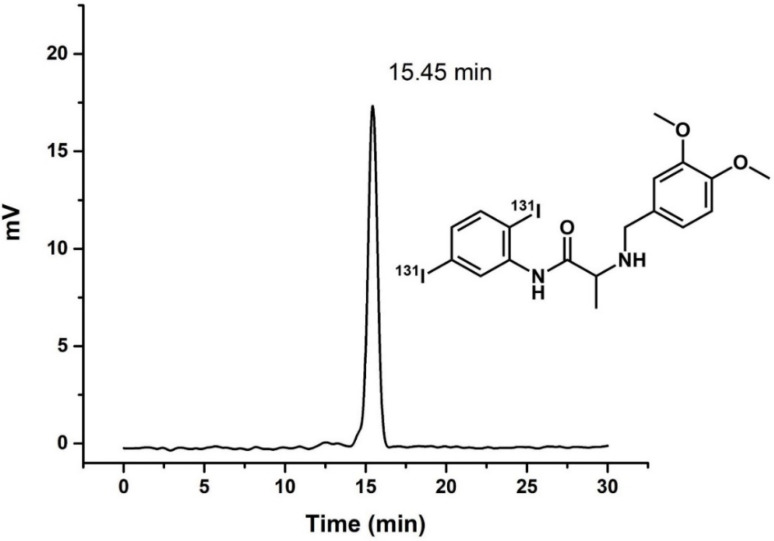
Reversed-phase radio-HPLC chromatogram of the ^131^I-labeled C19 compound. General scheme of I-C19 structure labeled with ^131^I (top-right).

**Figure 4 molecules-27-05446-f004:**
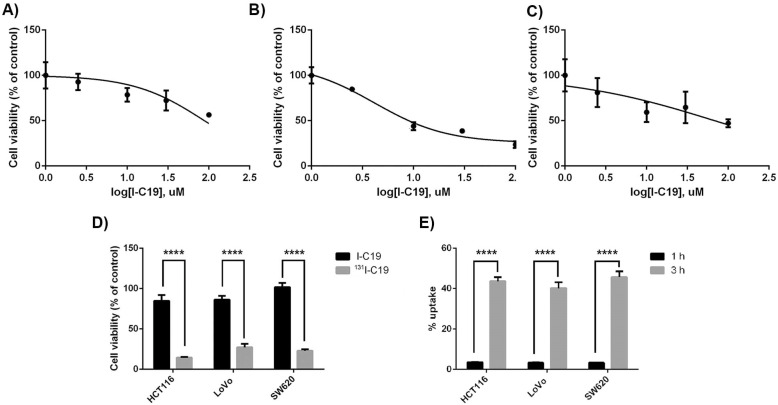
The colorectal cancer cell lines (**A**) HCT116; (**B**) LoVo; (**C**) SW620 treated with the indicated concentrations of I-C19 for 72 h (*p* < 0.05); (**D**) treatment with ^131^I-C19 at a concentration of 1.8, 6.8 and 8.8 µM in LoVo, SW620 and HCT116, respectively, significantly reduced the viability of the three colorectal cancer cell lines in comparison with unlabeled I-C19; (**E**) ^131^I-C19 uptake experiments. After incubation for 1, 3 and 5 h, ^131^I-C19 uptake by cancer cells significantly increased. n = 3; **** *p* < 0.0001.

**Figure 5 molecules-27-05446-f005:**
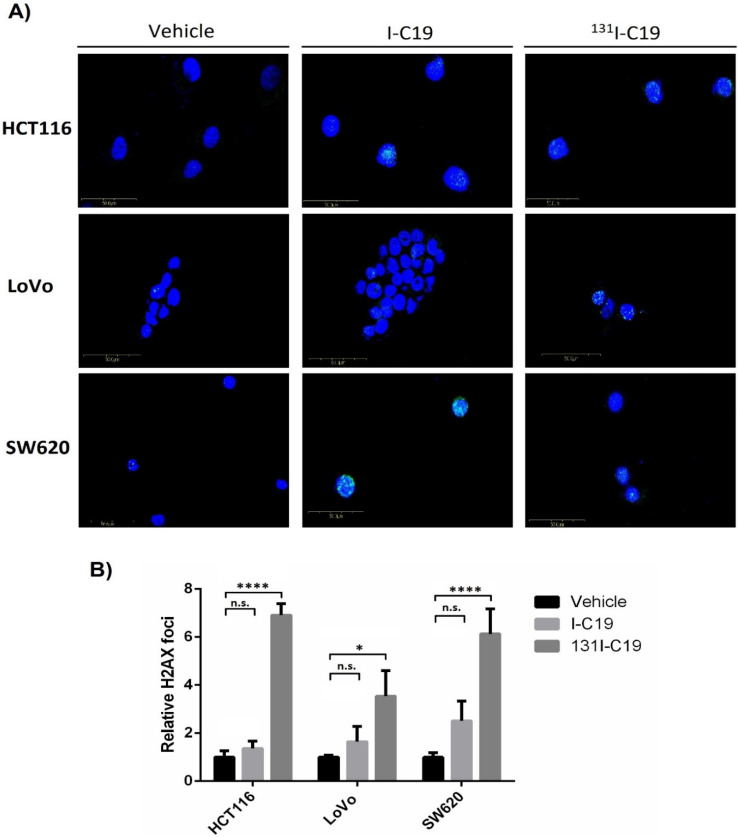
(**A**) Representative images of the confocal microscope showing γ-H2AX foci in the nucleus (DAPI) of colorectal cancer cells treated with the vehicle and the IC_50_ of I-C19 (a tenth of the IC_50_ concentrations of each cell line) for 3 h; (**B**) relative γ-H2AX foci of cancer cells treated with I-C19, ^131^I-C19 and control cells. Data are presented as the mean ± SEM from three independent experiments. * *p* < 0.1; **** *p* < 0.0001.

**Figure 6 molecules-27-05446-f006:**
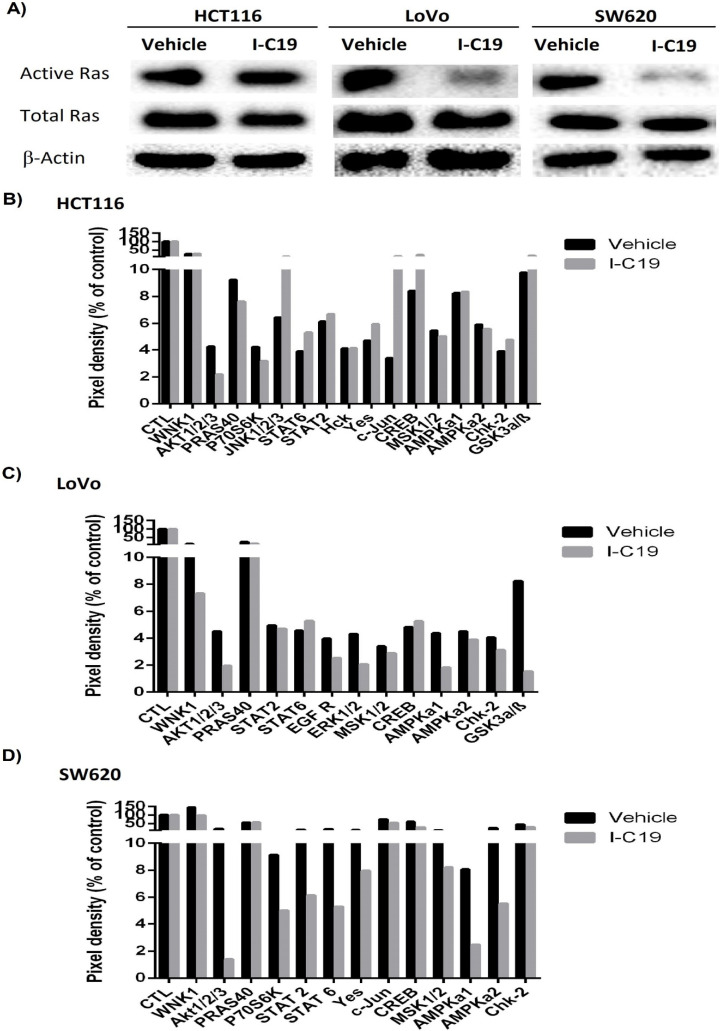
(**A**) RAS-GTP pull-down assay of three colorectal cancer cell lines treated with the vehicle and the IC_50_ of I-C19 for 24 h, and the total Ras and β-actin were the loading controls (cropping blots); (**B**) human phospho-kinase array of the total protein extract of the colorectal cancer HCT116 cell line treated with the IC_50_ of I-C19 for 24 h. (**C**) human phospho-kinase array of the total protein extract of the colorectal cancer LoVo cell line treated with the IC_50_ of I-C19 for 24 h. (**D**) human phospho-kinase array of the total protein extract of the colorectal cancer SW620 cell line treated with the IC_50_ of I-C19 for 24 h. Pixel density—% of control.

**Figure 7 molecules-27-05446-f007:**
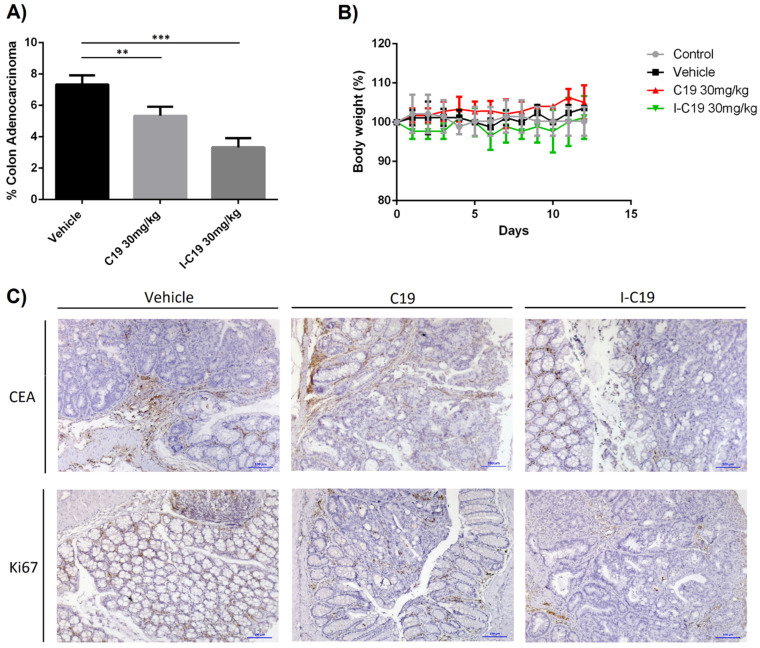
(**A**) The effect of I-C19 treatment on tumor growth of AOM/DSS-treated mice, showing the percentage of colon adenocarcinoma among the mice. The mice were given C19 (30 mg/kg) or I-C19 (30 mg/kg) intraperitoneally every day for 12 days. (**B**) Body weight percentage of AOM/DSS-treated mice, measured every day during treatment. (**C**) Representative pictures of IHC staining for CEA and Ki67 in colorectal tissues. Data are presented as the mean ± SEM from independent experiments ** *p* < 0.01, *** *p* < 0.001.

**Figure 8 molecules-27-05446-f008:**
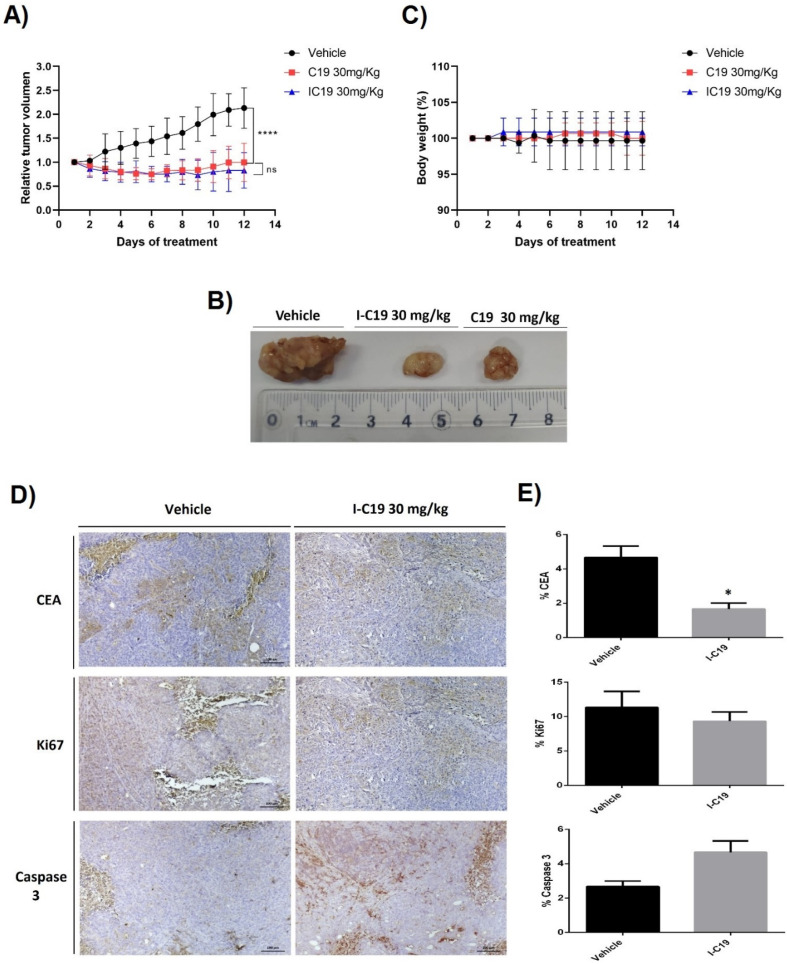
(**A**) I-C19 was administered every day for 12 days at a concentration of 30 mg/kg. Tumor volume was measured daily and estimated according to the formula: V =(½ d)(D^2^); **** *p* < 0.001. (**B**) Representative images of tumor volume after dissection. (**C**) Body weight percentage for tumor-bearing nude mice measured every day during treatment. **(D)** Representative images of IHC staining for CEA, Ki67 and cleaved caspase-3 in the xenograft tumor tissues. (**E**) Graphs showing the percentage of neoplastic tissue positive to stain; * *p* < 0.1.

**Figure 9 molecules-27-05446-f009:**
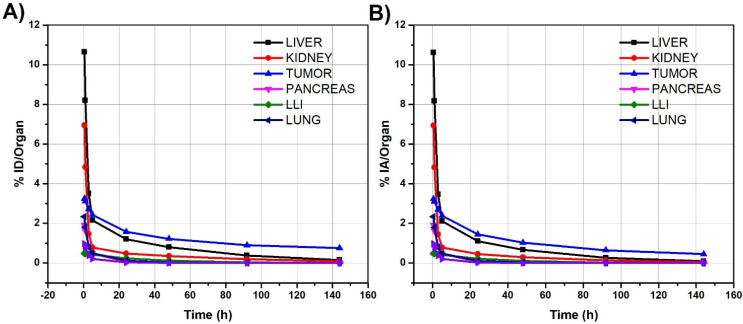
Biokinetic profile of (**A**) I-C19 [$q_{h}\left(t\right)$] and (**B**) ^131^I-C19 [$A_{h}\left(t\right)$ ] in Nu/Nu mice with LoVo xenograft tumors after intraperitoneal administration.

**Figure 10 molecules-27-05446-f010:**
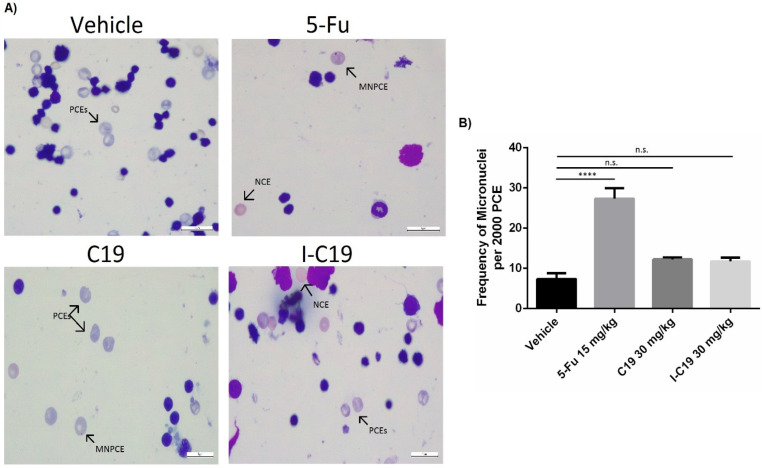
(**A**) Representative images of bone marrow cells from mice 24 h after a single injection of 15 mg/kg of 5-Fu and 30 mg/kg of C19 and I-C19. The arrows show the polychromatic erythrocytes (PCEs), micronuclei of polychromatic erythrocytes (MNPCEs) and normochromic erythrocytes (NCEs). (**B**) Values of micronuclei counts in the bone marrow cells. Data are presented as the mean ± SEM from four independent experiments **** *p* < 0.0001; n.s., non-significant.

**Table 1 molecules-27-05446-t001:** Binding free energy components of the protein–ligand complexes (kcal/mol).

Protein–Ligand Bound KRas4B–PDE6δ Complex							
System	ΔE_vdw_	ΔE_ele_	ΔG_ele,sol_	ΔG_npol,sol_	ΔE_non-polar_	ΔE_polar_	ΔG_bind_
K-Ras4B–PDEδ–C19	−137.38 (7.29)	−1236.33 (65.30)	1304.69 (65.80)	−18.79 (0.88)	−156.17	68.36	−87.81 (7.0)
KRas4B^G13D^–PDE6δ–C19	−133.04 (8.23)	−1443.00 (74.28)	1499.25 (72.56)	−19.03 (0.98)	−152.07	56.25	−95.82 (8.0)
KRas4B–PDE6δ–I-C19	−150.23 (8.50)	−1369.00 (79.80)	1438.07 (75.23)	−21.72 (0.90)	−171.95	69.07	−102.88 (10.40)
KRas4B^G13D^–PDE6δ–I-C19	−159.09 (12.50)	−1406.58 (87.70)	1440.02 (82.58)	−23.58 (1.46)	−182.67	33.44	−149.23 (18.80)

**Table 2 molecules-27-05446-t002:** Radiation absorbed doses produced by the uptake and internalized activity of 131I-C19 to the HCT116, LoVo and SW620 cancer cell nuclei.

Cell Line	Cell Surface Total Nuclear Transformations (Bq^.^s)	Cytoplasm Total Nuclear Transformations (Bq^.^s)	Absorbed Dose (Gy)	
			D_N__←CS_	D_N__←Cy_
HCT116	2023	47	0.18	0.01
LoVo	1609	918	0.15	0.14
SW620	4248	367.2	0.38	0.05

**Table 3 molecules-27-05446-t003:** Total nuclear transformations and radiation absorbed dose produced by 5 MBq of ^131^I-C19 to the organs and tumor (intraperitoneal administration).

Organ	Total Nuclear Transformation $\;N=\int_{0}^{\infty }A_{h}\left(t\right)dt$ (MBq.s)	Absorbed Dose (Gy)
Liver	3816	0.34
Pancreas	399	0.18
Kidney	1796	0.12
Colon (Large intestine)	504	0.14
Lung	421	0.40
Spleen	151	0.19
Tumor	7020	3.55

## Data Availability

Not applicable.

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
