# Peer review of "131I-C19 Iodide Radioisotope and Synthetic I-C19 Compounds as K-Ras4B–PDE6δ Inhibitors: A Novel Approach against Colorectal Cancer—Biological Characterization, Biokinetics and Dosimetry"

_molecules, 2022, doi:10.3390/molecules27175446_

Round 1

Reviewer 1 Report

In my point of view as a computational medicinal chemist,  the research presented in the manuscript is interesting and could be published in Molecules. However, there are some major issues that need to be revised in the manuscript before it can go for publication. The following are those issues:

1. The English used in the manuscript should be increased and some sentences need to be re-written to make the manuscript more understandable. Especially, some sentences in the Introduction section. For example: "Activating mutations in K-Ras4B mainly consist of oncogenic substitutions in G12 and G13 residues, which impair intrinsic and GAP-mediated GTP hydrolysis." (lines 51-53).

2. The Abstract section should be reformatted to match the Abstract format for Molecules.

3. Please provide some contexts on the correlation of the molecular modeling studies, especially the selection of the complex to the in vivo data. It would assist the reader to understand the story. I need to dig back into the cited references to understand this context.  

4. Please provide the statistical significance of Figures 2A-C. I think that we could not derive the IC50 values based on the data.

Author Response

Dr. Yu-shan Wu

Dr. Guor-Jien Wei 

Guest Editors

Molecules

Manuscript ID: molecules-1841823

131I-C19 iodide radioisotope and synthetic I-C19 compounds as K-Ras4B–PDE6δ inhibitors: a novel approach against colorectal cancer—Biological characterization, biokinetics and dosimetry”

Dear Professors,

Please find attached the answers, point-by-point, to the reviewer's comments.

REVIEWER 1

  1. The English used in the manuscript should be increased and some sentences need to be re-written to make the manuscript more understandable. Especially, some sentences in the Introduction section. For example: "Activating mutations in K-Ras4B mainly consist of oncogenic substitutions in G12 and G13 residues, which impair intrinsic and GAP-mediated GTP hydrolysis." (Lines 51-53).

Answer

The manuscript molecules-1841823 has undergone English language editing by MDPI (see attached: English editing certificate). Therefore, the sentence mentioned by the reviewer was corrected.

“Activating mutations in K-Ras4B mainly consist of substitutions in G12 and G13 residues, which impair intrinsic GTP hydrolysis mediated by GTP activating protein (GAP).”

  1. The Abstract section should be reformatted to match the Abstract format for Molecules.

Answer

Thank you for the observation. The Abstract was modified in agreement with the format for molecules.

  1. Please provide some contexts on the correlation of the molecular modeling studies, especially the selection of the complex to the in vivo data. It would assist the reader to understand the story. I need to dig back into the cited references to understand this context.  

Answer

The text was modified to explain the selection of the complex and the correlation of the molecular modeling studies as follows:

Abstract: Lines: 24-31. The text was rewritten as follows:

In 40-50% of colorectal cancer (CRC) cases, K-Ras gene mutations occur, which induce the ex-pression of the K-Ras4B oncogenic isoform. K-Ras4B is transported by phosphodiesterase-6δ (PDE6δ) to the plasma membrane, where the K Ras4B–PDE6δ complex dissociates and K-Ras4B, coupled to the plasma membrane, activates signaling pathways that favor cancer aggressiveness. Thus, the inhibition of the K-Ras4B– PDE6δ dissociation using specific small molecules could be a new strategy for the treatment of patients with CRC. This research aimed to perform a preclinical proof-of-concept and the therapeutic potential evaluation of the synthetic I-C19 and 131I-C19 compounds as inhibitors of the K-Ras4B–PDE6δ dissociation.”

Introduction Section . Lines 67-75.  The following text was added:

“It has been demonstrated that the inhibition of the K-Ras4B and PDE6δ association impairs the oncogenic K-Ras signaling [5]. However, stabilization of the K-Ras4B–PDE6δ complex has been recently proposed by our group as a novel strategy to reduce the release of K-Ras4B to the plasma membrane, thus inhibiting K-Ras4B signaling [6, 7].”

“In this approach, which to our knowledge has not previously been reported by another research group, a synthetic small molecule is used to specifically bind to the interaction site between K-Ras4B and PDE6δ mimicking the effect of a “staple”, thus preventing the K-Ras4B–PDE6δ dissociation. Therefore, the inhibitory effect of C19 causes a decrease in the concentration of K-Ras4B in the plasma membrane and blocks the activation of the mitogen-activated protein kinase (MAPK) signaling pathway.”

Results Section 2.1. Lines 86-90. The following text was added:

“In order to evaluate the binding affinity as well as the anchorage sites of the C19 and I-C19 compounds on the wild-type (K-Ras4BWT–PDE6δ) and mutant (K-Ras4BG13D–PDE6δ) system, docking and molecular dynamics (MD) were performed as previously reported [7].”

Results Section 2.3. Lines 172-177. The following text was added:

“This result allowed us to corroborates that the I-C19 compound has a higher affinity than the C19 compound by inhibiting the viability of K-Ras4B-dependent cells at a lower concentration [7]. As shown in the calculation of the binding free energy, I-C19 presented a higher binding energy than C19 (-95.82 vs -149.23 kcal/mol for C19 and I-C19 in the molecular complex K-Ras4B–PDE6δ, respectively) (Table 1).“

Results Section 2.6. Lines 247-251. The following text was added:

“These results correlate with the results of the binding free energy calculations in which the compound I-C19 presented a ΔG binding of -95.82 kcal/mol when it interacted with the molecular complex K-Ras4B–PDE6δ; however, when the compound I-C19 interacted with the complex K-Ras4BbG13D–PDE6δ (KRAS mutated), the ΔG binding increased (-149 kcal/mol) (Table 1).”

  1. Please provide the statistical significance of Figures 2A-C. I think that we could not derive the IC50 values based on the data.

Answer

Statistical significances of figures 4A-C were p<0.05, and it was noted on figure 4 description.

We perform a Nonlinear fit of transform of log[inhibitor] vs response - Variable slope and best-fit values was IC50 18.39, 88.24 and 68.28 for LoVo, HCT116 and SW620 cell lines respectively. The IC50 estimated a LogIC50 standard error of 0.1102, 0.09411 and 0.2308 and 95% confidence intervals (α=0.05) of 1.027 to 1.503, 1.742 to 2.149 and 1.336 to 2.333 for LoVo, HCT116 and SW620 cell lines respectively. The IC50 values for I-C19 were calculated using the GraphPad Prism.

We change graphics (Figure 4A-C) to Sigmoidal (X is concentration) for a better fitting (R2= 0.95).

Reviewer 2 Report

The authors present a complete characterization of a compound and its iodide radioisotope as K-Ras4B/PDEd inhibitors. The manuscript describes a very interesting work and will be useful to those working in anticancer drug design and specially in colorectal cancer. However, there are minor corrections before its acceptance for publication.

1.- It would be desirable to include the trajectories of the molecular dynamics simulations with some additional analysis such as RMSD, RMSF, and RG.

2.- I the same context, nothing is discussed about the interactions made by the compounds, only the interaction type is mentioned. 

3.- Please include the model used to calculate IC50 in cell viability. Additionally, the highest concentration used to this end was very close to the 50%, why do not use higher concentrations to get a better fitting and a more precise IC50 estimation?

4.- Supplemental material is mentioned in the text, however, the file was not included.

Author Response

REVIEWER 2

The authors present a complete characterization of a compound and its iodide radioisotope as K-Ras4B/PDEd inhibitors. The manuscript describes a very interesting work and will be useful to those working in anticancer drug design and specially in colorectal cancer. However, there are minor corrections before its acceptance for publication.

1.- It would be desirable to include the trajectories of the molecular dynamic’s simulations with some additional analysis such as RMSD, RMSF, and RG.

Answer

The trajectories of the molecular dynamic’s simulations with some additional analysis such as RMSD and RG were included as the reviewer suggested. Figure 1 was added as follows:

“Figure 1. (A) RMSD and (B) Rg values of the complexes between KRas4BWT–PDE6δ or KRas4BG13D–PDE6δ with C19 or I-C19. RMSD (left) values of KRas4BWT–PDE6δ-C19 (black line) KRas4BG13D–PDE6δ-C19 (red line), KRas4BWT–PDE6δ-I-C19 (blue line), and KRas4BG13D–PDE6δ-I-C19 (magenta line). Rg (right) values of KRas4BWT–PDE6δ-C19 (black line) KRas4BG13D–PDE6δ-C19 (red line), KRas4BWT–PDE6δ-I-C19 (blue line), and KRas4BG13D–PDE6δ-I-C19 (magenta line).”

Result Section 2.1. Lines 90-96.  The following text was added:

“Analysis of the root-means-squared-deviation (RMSD) showed that K-Ras4BWTPDE6δ-C19, K-Ras4BG13DPDE6δ-C19, K-Ras4BWTPDE6δ-I-C19, K-Ras4BG13DPDE6δ-I-C19 complexes reached stable values between 40 to 60 ns, with values oscillating between 1.9 to 3.2 Å (Figure 1A). Evaluation of the radius of gyration (Rg) also showed equilibration at similar simulation times, with values about 22.5 Å (Figure 1B).”

2.- I the same context, nothing is discussed about the interactions made by the compounds, only the interaction type is mentioned. 

Answer

In order to clarify this point, the following paragraphs were added:

Results Section 1. Lines 67-75:

“In this approach, which to our knowledge has not previously been reported by another research group, a synthetic small molecule is used to specifically bind to the interaction site between K-Ras4B and PDE6δ mimicking the effect of a “staple”, thus preventing the K-Ras4B–PDE6δ dissociation. Therefore, the inhibitory effect of C19 causes a decrease in the concentration of K-Ras4B in the plasma membrane and blocks the activation of the mitogen-activated protein kinase (MAPK) signaling pathway.”

Results Section 2.1. Lines 86-90:

“In order to evaluate the binding affinity as well as the anchorage sites of the C19 and I-C19 compounds on the wild-type (K-Ras4BWTPDE6δ) and mutant (K-Ras4BG13DPDE6δ) system, docking and molecular dynamics (MD) were performed as previously reported [7].”

Section 2.3. Lines 172-177. This is how it is described:

“This result allowed us to corroborates that the I-C19 compound has a higher affinity than the C19 compound by inhibiting the viability of K-Ras4B-dependent cells at a lower concentration [7]. As shown in the calculation of the binding free energy, I-C19 presented a higher binding energy than C19 (-95.82 vs -149.23 kcal/mol for C19 and I-C19 in the molecular complex K-Ras4B–PDE6δ, respectively) (Table 1).“

Section 2.6. Lines 247-251. This is how it is described:

“These results correlate with the results of the binding free energy calculations in which the compound I-C19 presented a ΔG binding of -95.82 kcal/mol when it interacted with the molecular complex K-Ras4B–PDE6δ; however, when the compound I-C19 interacted with the complex K-Ras4BbG13D–PDE6δ (KRAS mutated), the ΔG binding increased (-149 kcal/mol) (Table 1).”

3.- Please include the model used to calculate IC50 in cell viability. Additionally, the highest concentration used to this end was very close to the 50%, why do not use higher concentrations to get a better fitting and a more precise IC50 estimation?

Answer

As previously we reported, the maximum concentration to avoid crystallization and nonspecific results was 100µM (Reference 7 in manuscript). Therefore, the used concentration for IC50 determination was chosen from 0 to 100 mM, please be aware that the x-axis is on log scale. Therefore, the following phrase was modified, section 2.3, Lines 169-170:

“The HCT116 and SW620 cancer cell lines did not produce cytotoxic effects in high concentrations (i.e., more than 100 µM; data not shown).”  

We perform a Nonlinear fit of transform of log[inhibitor] vs response - Variable slope and best-fit values was IC50 18.39, 88.24 and 68.28 for LoVo, HCT116 and SW620 cell lines respectively. The IC50 estimated a LogIC50 standard error of 0.1102, 0.09411 and 0.2308 and 95% confidence intervals (α=0.05) of 1.027 to 1.503, 1.742 to 2.149 and 1.336 to 2.333 for LoVo, HCT116 and SW620 cell lines respectively. The IC50 values for I-C19 were calculated using the GraphPad Prism, as described in the manuscript.

We change graphics (Figure 4A-C) to Sigmoidal (X is concentration) for a better fitting (R2= 0.97).

4.- Supplemental material is mentioned in the text, however, the file was not included.

Answer

There are not supplementary images, they were including as part of the article. Text was corrected on section 2.3. Line 153 and 156.

Reviewer 3 Report

Please, see attached file

Author Response

August 13, 2022

Dr. Yu-shan Wu

Dr. Guor-Jien Wei 

Guest Editors

Molecules

Manuscript ID: molecules-1841823

131I-C19 iodide radioisotope and synthetic I-C19 compounds as K-Ras4B–PDE6δ inhibitors: a novel approach against colorectal cancer—Biological characterization, biokinetics and dosimetry”

Dear Professors,

Please find attached the answers, point-by-point, to the reviewer's comments.

REVIEWER 3

The article by Cruz-Nova et al presents a thorough characterization of 131I-CA19 as a therapeutic agent against colorectal cancer. The characterization is extremely complete and includes synthesis, in vitro cytotoxicity, biokinetics, in vivo response and dosimetry.

I am mostly familiar with biokinetics, dosimetry and radiobiological response, so this review will be mostly focused on these points.

I have some serious concerns on the dosimetry and radiobiological response that I hope the authors can clarify before publication.

Biokinetic modelling and dosimetry

  1. In section 4.18 the authors state that activities in several organs were measured at 1, 3 and 24h, and then introduced in Olinda to fit the time-activity-curves for each of those organs. The authors should present these experimental data in a figure, as well as the fit used to obtain A(t) (and so the dose). This figure can replace Table 3, which is not easy to interpret, and Figure 7(B).

My fear is that these time points may not be adequate to obtain a good fit of the A(t) curve, because the physical half-life of 131-I is much larger (~8 days) and the biological half-life might be larger. From Figure 7(B) it seems that this may not be the case, but a new figure showing A(t) vs t and the fits would be more clear.

Regarding Table 3, what are the units of q(t), A(t) and t in the fits that are reported?. What are the units of integral(A(t))? (certainly, it cannot be total disintegrations, because that number would be much larger). What do q(t) and A(t) represent?. I’m afraid that the meaning of “biological behaviour” and “biological behaviour corrected by decay” is not totally clear.

Answer

As recommended the reviewer, Figure 7B and Table 3 (biokinetic model) were substituted by the biokinetic profile figure of compound, I-C19 [qh(t)] and 131I-C19 [Ah(t)].

Methods were modified in the Methodology section 4.16 y 4.17 to clarify the main concerns of the reviewer as follows (please, see the manuscript where equations are written adequately using the word equation editor):

  • “Biodistribution and biokinetic models

131I-C19 was intraperitoneally injected (200 µL, 5 MBq (135 µCi/mice)) into nu/nu mice. The mice (n = 3) were sacrificed at 0.5, 1, 3, 5, 24, 48, 92 and 144 h post-injection. Whole tumor, lung, liver, pancreas, spleen, kidney, large intestine (colon) and blood were rinsed and placed into preweighed plastic test tubes. The activity was determined in a NaI(Tl) detector (Canberra) along with three 0.5 mL aliquots of the diluted standard, representing 100% of the injected activity and used for radioactive decay correction. Mean activity values in each organ were correlated with those of the standard (representing the initial injected activity) to obtain the percentage of injected dose (% ID) at different times. The % ID values of each organ or tumor were input into the Olinda/EXM 2.0 software and adjusted to exponential functions (first-order kinetics) to obtain biokinetic models corrected by radioactive decay (qh(t))  (Eq. 2), which corresponded to the I-C19 compound. Likewise, the Ah(t) functions, which corresponded to the 131I-C19 biokinetic models, were obtained considering lamdaeff = lambdaB+ lambaR, where the biological constant is represented by λB and the 131Iradioactive decay constant is lambdaR (Eq. 3).

qh(t)= Ah(t) x exp(lambdaR) = B x exp(lambdaBb)t + C x exp(lambdaBc)t + D x exp(lambdaBd)t        (eq. 2)

Ah(t)= B x exp(lambdaBb + lambdaR)t + C x exp(lambdaBc + lambdaR)t + D x exp(lambdaBd  + lambdaR)t        (eq. 3)

4.17 Radiation absorbed dose estimation

The mathematical integration of  functions yielded the total nuclear transformations (MBq.s) that occurred in each source organ (Eq. 4).

(4)

The absorbed doses delivered by 131I-C19 to selected organs were estimated based on the following equation 5:

(5)

where  is the mean absorbed dose to the target organ from a source organ;  is a dose factor that considers the fraction of absorbed energy for each nuclear emission of iodine-131 in the different geometries and chemical compositions of the organs. Iodine-131 DF values (mGy/MBq.s) for a 25 g mouse model were obtained from the Olinda code, version 2.2. For the tumor dose calculation, the sphere model was used considering an average tumor mass of 0.1 g (Olinda 1.2 software).

  1. In section 4.18 the authors state that they use Olinda to obtain the time-activity-curve A(t) corrected by decay, and then they use this curve to obtain the total number of disintegrations as:

Nsource=∫0 ∞ A(t)dt

This computation might not be correct. If A(t) is the activity corrected by decay, the calculation of the number of disintegrations should include the physical decay of the source, i.e.:

Nsource=∫0 ∞ A(t)exp(−λ t)dt

Please, check this equation. Either A(t) or the integral should incorporate physical decay in order to properly compute the number of disintegrations (and the dose).

Also, please notice a missing “=” sign in equation (2), and missing arrows in equation (3) (“target←source”).

Answer

In the Methodology section 4.17. Lines 668-675, It was clarified to point out that A(t) includes physical decay. Therefore, equation 2 was corrected.

Typo on Equation 4 was corrected. (please, see the manuscript where equations are written adequately using the word equation editor):

4.17 Radiation absorbed dose estimation

The mathematical integration of  functions yielded the total nuclear transformations (MBq.s) that occurred in each source organ (Eq. 4).

(4)

The absorbed doses delivered by 131I-C19 to selected organs were estimated based on the following equation 5:

(5)

where  is the mean absorbed dose to the target organ from a source organ;  is a dose factor that considers the fraction of absorbed energy for each nuclear emission of iodine-131 in the different geometries and chemical compositions of the organs. Iodine-131 DF values (mGy/MBq.s) for a 25 g mouse model were obtained from the Olinda code, version 2.2. For the tumor dose calculation, the sphere model was used considering an average tumor mass of 0.1 g (Olinda 1.2 software).

  1. Regarding the dosimetry for the in vitro experiments, I have several doubts about how the absorbed doses were computed:

- the authors refer to S values taken from Guddu et al for 131-I. However, Guddu’s paper only reports S values for cell radius = 10 um and nucleus radius = 9, 8, 7 um. Did the authors extrapolate these data to get S values for 10/6 (Lovo and SW620 cells) and 10/5 (HCT116)?.

Answer

The S values for HCT116 (cell surface=6.09E-05; cytoplasm=1.70E-04), LoVo and SW620 (cell surface=6.27E-05 cytoplasm=1.08E-04) were taken from “Goddu SMBTF: MIRD cellular S. values: self-absorbed dose per unit cumulated activity for selected radionuclides and monoenergetic electron and alpha particle emitters incorporated into different cell compartments. Reston, VA: Society of Nuclear Medicine; 2003, pag 96”. In this reference, S values (I-131) for cell radius = 10 µm and nucleus radius = 5, 6, 7, 8 and 9 micrometers  are reported.

The reference was corrected in the manuscript, section 4.8. Lines 579-581. Thanks for the clarification.

“The dose factor geometries were obtained from S values [32] (HCT116 cell radius = 10 mm, nucleus radius = 5mm ; LoVo and SW620 cell radius = 10 mm, nucleus radius = 6 mm).”

- I am guessing that in order to obtain NCy and NCS the authors use the information obtained from may lead to a systematic error in the calculation of the uptake/internalization activity. Did the authors consider this? the internalization and uptake experiments performed at 3h. However, uptake/internalization is most likely dynamic during the 72 h of the in vitro experiments, and a single time point evaluation at 3h

Answer

Uptake (cell surface) and internalization (cytoplasm) experiments were carried out at 1, 3 and 5 h (data available). However, to point out significant results only 1 and 3 h were showed in Figure 2E.

Thanks for your comment, the clarification was made on section 4.5 (cell uptake). Line 533:

“The uptake (%) (plasmatic membrane) and internalization (%) (cytoplasm) were used to obtain the radiation absorbed dose.”

- the authors report very different absorbed doses for the HCT116 (1.88 Gy), LoVo (2.88 Gy) and SW620 (0.43 Gy) cells. What is the origin of such differences?. Uptake values (at 3h) are very similar between cell types (from 40% to 45%), and S-values should not be very different (especially for Lovo and SW620 cells, for which the same cell radius and nucleus radius are assumed). What is the cause of that x7 dose difference?

- Table 2 tries to explain the dosimetry calculation, yet to this reviewer it is not very clear. From Table 2 one may infer that doses are so different because uptake and internalization activities are also quite different. Why are uptake values in table 2 so different between different cells when figure 2E shows very similar uptakes?

Answer

The origin of different absorbed doses is the different number of nuclear transformations in each biokinetic model obtained from Olinda 2.0 software. The model was Integrated from 0 to infinite and not from 0 to 72 h as previously written. Titles in table 2 were modified to make the results clearer.

Table 2. Radiation absorbed doses produced by the uptake and internalized activity of 131I-C19 to the HCT116, LoVo and SW620 cancer cell nuclei.

Cell Line

Cell Surface

Total nuclear transformations

(Bq.s)

Cytoplasm  

Total nuclear transformations

(Bq.s)

Absorbed dose

(Gy)

  DNCS         DNCy

HCT116

20232

471.6

1.77

0.11

LoVo

16092

9180

1.45

1.43

SW620

4248

367.2

0.38

0.05

- For the calculation of in vitro doses during the 72h experiments the authors seem to be considering just uptake (NCS) and internalization NCy ) activities. However, extracellular activity will also contribute to absorbed doses in the cell because the range of the radiation emitted by 131-I is much larger than the cell size. As far as I understand, cells are not washed up to remove extracellular activity. Is this contribution taken into account?

Answer

Cells were washed two times with PBS and treated with 500 µL of glycine buffer, as indicated in section 4.5. Lines 537-539. Therefore, extracellular activity did not contribute to absorbed doses.

“After treatment, two washes (1 mL of ice-cold PBS) were performed. For calculation of the cell uptake, the cells in each well were treated (at room temperature for 5 min) with 500 mL of glycine buffer (50 mM, pH 2.8).”

- In section 4.7 it is stated that cells were exposed to 8.8, 1.8 and 6.8 uM of 131I-C19 (HCT116, LoVo and SW620). However, in section 2.3 it is stated that cells were exposed to 1.8 uM of 131IC19. Were cells exposed to different activities and therefore the differences in absorbed doses?

Answer

Cancer cells were exposed to different 131I-C19 molar concentration (10 times less than the IC50 in order to observe the effect on KRAS activation from I-C19) and the same activity (1.44 Bq/cell) as indicated in section 2.4. and section 4.7. Description of figure 4D and 4E was corrected.

“After treatment, two washes (1 mL of ice-cold PBS) were performed. For calculation of the cell uptake, the cells in each well were treated (at room temperature for 5 min) with 500 mL of glycine buffer (50 mM, pH 2.8).

For the assessment of dual therapy, beta radiation emission from 131I-labeled C19 was also evaluated. The mitochondrial dehydrogenase activity in living HCT116, LoVo and SW620 cells was considered to measure the contribution of 131I beta radiation to cell death, and the cell viability analysis was carried out after exposure to concentrations ten times lower than the IC50 obtained for each cancer cell line (8.8, 1.8 and 6.8 µM, respectively). 

Figure 4. The colorectal cancer cell lines (A) HCT116; (B) LoVo; (C) SW620 treated with the indicated concentrations of I-C19 for 72 h (p < 0.05); (D) treatment with 131I-C19 at a concentration of 1.8, 6.8 and 8.8 µM in LoVo, SW620 and HCT116, respectively, significantly reduced the viability of the three colorectal cancer cell lines in comparison with unlabeled I-C19; (E) 131I-C19 uptake experiments. After incubation for 1, 3 and 5 h, 131I-C19 uptake by cancer cells significantly increased. n = 3; **** p < 0.0001.  “ 

  1. In vivo experiments: the authors report the administration of 30 mg/Kg of C19 once daily for 12 days. It would be useful for the readers if the authors also report the injected activities of 131I-C19 in Figure 6 and section 4.14. In this way, absorbed doses in the tumor and organs could be easily calculated from injected activities and dose/activity factors reported in section 2.7.

In addition, it would be useful to report the total dose absorbed in the tumor (mean values) in Figure 6 A-C.

Answer

In vivo experiments for tumor growth inhibition attributed to K-ras4b signaling blockage were performed with non-radiolabeled I-C19 compound, as detailed in section 2.7. One perspective of this article is to probe the therapeutic effect of 131I-C19 on in vivo models.

Biokinetic modelling and dosimetry

  1. In vitro experiments

In figure 2D the authors report cell viability after exposure to 131I-C19. Surviving fractions seem to be below 20-25% after absorbing 1.88 Gy (HCT116), 2.88 Gy (Lovo) and 0.43 Gy (SW620), according to the dosimetry presented insection 2.3.

The last case is of especial interest, because it points out an extremely high radiosensitivity of SW620 cells. An analysis using the LQ model shows that a surviving fraction of ~25% after 0.43 Gy requires a radiosensitivity α~3 Gy-1 (α/β=10 Gy), well beyond the typical radiosensitivity of most cells. This calculation has been performed ignoring the protraction effect associated with a prolonged dose delivery (72 h), which would in fact lead to even higher radiosensitivities.

I have found a study reporting the radiosensitivity of SW620 to radiation (Endo et al Oncol Rep. 2018 Mar; 39(3): 1112–1118). They report a surviving fraction of ~0.74 at 1 Gy. I have performed a fit of their data to the LQ model, obtaining α~0.21 Gy-1 and α/β ~ 5Gy, which are quite reasonable values. Endo et al used X rays instead of 131-I: there may be a slight difference in the radiation quality of both beams, but not enough to explain the observed dfferences.

The observed dose/response effect seems to point out an underestimation of the absorbed dose.

Answer

The reviewer is omitting to consider the main objective of this research which was to demonstrate the synergistic effect of radiation and inhibition of KRAS activation on cell viability.

This paragraph was added in section 3. Lines 426-431.

“For example, Yang et al. (2021) describe the signaling pathway by which mutated KRAS can cause cells to acquire radioresistance. They demonstrated that KRAS mutations result in a more significant response to DNA damage and positive regulation of 53BP1 (a protein associated to dsDNA repair) with increased associated nonhomologous end-junction repair (NHEJ). Moreover, KRAS mutations lead to the activation of NRF2 antioxidant signaling to increase the transcription of the 53BP1 gene [23].”

Indeed, not only do we observe the effect of radiation dose on cell viability, together with the results of signaling pathways and pull down, we suggest that decreased viability is the effect of the combination of chemo-radiotherapy.

  1. In vivo experiments

For the in vivo experiments the authors use Lovo cells. It is certainly interesting that while the in vitro experiments show a large difference between cell viability after exposure to C19 and 131I- C19, this is not the case in the in vivo experiments, where the response to C19 and 131I-C19 is very similar.

Certainly, in vivo and in vitro responses can be very different. For example, it may happen that the tumor shrinking time is much longer than 12 days, and differences between C19/131I-C19 will become significant at later times.

Providing the absorbed dose in the tumor in the in vivo experiments would help to understand this response (or lack thereof) to 131I-C19.

If the authors cannot show a different in vivo response (volume dynamics) to C19 and 131I-C19 they should be cautious in the conclusions about improved response to 131I-C19.

Answer

The in vivo experiments were only performed with the compounds C19 and I-C19 but no 131I-C19. The results from section 2.7 and figure 7 represent the therapeutic effect of C19 and I-C19 and as you mentioned there is no significant difference in the in vivo effect of both compounds. The advantage of I-C19 is that it can be radiolabel with iodine 131 radioisotope for biokinetics, dosimetry and combination of chemo-radiotherapy.

Other comments:

- Please, check the references. There seems to be a problem with the order in which they are cited. For example, in line 553 “reported by Guddu and Budinger [30]”, but Guddu and Budinger (and Rao) is reference 31. In line 482-83, the reference to Amber16 suite should be [26], not [27]. There are more instances of this problem.

The references were revised and corrected

- The authors refer to two supplementary figures in the text. This referee has not had access to these supplementary figures for this review. Please, include them in the resubmission

Answer

There are not supplementary images, they were including as part of the article. Text was corrected on section 2.2. Line 153 and 156.

Round 2

Reviewer 1 Report

The manuscript in the current version could go further for publication in Molecules.

Author Response

Dr. Yu-shan Wu

Dr. Guor-Jien Wei 

Guest Editors

Molecules

Manuscript ID: molecules-1841823

131I-C19 iodide radioisotope and synthetic I-C19 compounds as K-Ras4B–PDE6δ inhibitors: a novel approach against colorectal cancer—Biological characterization, biokinetics and dosimetry”

Dear Professors,

Please find attached the answers, point-by-point, to the reviewer comments.

REVIEWER

The article by Cruz-Nova et al presents a thorough characterization of 131I-C19 as a therapeutic agent against colorectal cancer. The authors have addressed several of the concerns raised by this referee, especially regarding the in vivo data. However, I still have some concerns regarding the in vitro dosimetry and cell viability. For SW620 cells the authors report dose/viability data that point out to extremely radiosensitive cells, well beyond anything previously measured. I have concerns regarding this data (specifically, the dose values). I’d like the authors to perform one final check of the in-vitro dose calculations, to provide a better description of the dosimetry, and if they fully trust their calculation, to include a paragraph describing the extreme radiosensitivity of these cells and linking it to previous studies.

Answer:

In this work, the intention of the in vitro cell studies was not to obtain cytotoxic effects due to an ablative radiation dose to cancer cells produced by I-131, which is a very well-known issue; nor demonstrate that cells are extremely radiosensitive. Our aim was to evaluate the cytotoxic effect of the I-C19 in radiosensitized cells (synergistic effect between I-C19 and I-131 radiation); that’s why intentionally we prepared 131I-C-19 with a very low specific activity. What we demonstrated is that approximate 0.2 Gy effectively potentiate the cytotoxicity of I-C19 which is the contribution of this research. The inhibition of K-ras4B and its signaling pathways is directly proportional to the radiosensitizing effect on cancer cells [1]. Moreover, radiation absorbed doses at 0.2 Gy are sufficient to induce deterministic effects in human cells, for example, the changes observed in blood cells at 0.2 Gy. 

In order to clarify this point, the following paragraph was added in line 193-198:

“The inhibition of K-ras4B and its signaling pathways is directly proportional to the radiosensitizing effect on cancer cells [1]. Absorbed radiation doses at 0.2 Gy are enough to induce deterministic effects in human cells, for example, the phenotypic changes observed in blood cells. Therefore, our results demonstrated that a dose of approximately 0.2 Gy is sufficient to effectively potentiate I-C19 cytotoxicity (synergistic effect between I-C19 and I-131 radiation).”

[1] Gurtner K, Kryzmien Z, Koi L, Wang M, Benes CH, Hering S, Willers H, Baumann M, Krause M. Radioresistance of KRAS/TP53-mutated lung cancer can be overcome by radiation dose escalation or EGFR tyrosine kinase inhibition in vivo. Int J Cancer. 2020 Jul 15;147(2):472-477. doi: 10.1002/ijc.32598.

In table 2 the authors report surface and cytoplasm integrated activities for each type of cell, which then they convert to doses by using the coefficients provided in reference 33. I’m guessing that integrated activities are calculated by integrating the time-activity-curves obtained in the biokinetic study reported in section 2.4 (lines 178-184). In that paragraph the authors report a ~40% membrane uptake at 3h, which decreases to ~2% at 5h for HCT116/LoVo cells, but only to 25% for SW620 cells. This seems to point out a slower membrane activity clearance for SW620, which one would expect to lead to larger integrated activities.

Yet, the integrated membrane activity for SW620 cells is 5 times lower than for HCT cells.

There seems to be an inconsistency between biokinetic data presented in lines 178-184 and Ncs / Ncy values presented in Table 2. If the calculation of the integrated activities is not performed as guessed, the authors should provide an explanation of how it is done.

Answer:

Thank you for the observation, the mistake was corrected. The correction in the placement of the period in HCT116 and LoVo cells were done (Table 2); therefore, the doses were also corrected (Table 2). However, the total doses remained at approximately 0.2 Gy in all cases.

The authors should acknowledge that if their dosimetry calculations are correct, a cell viability of 25% after irradiation with 0.43 Gy is quite astonishing (see e.g. the reference provided). If the dosimetry is correct, the authors should mention the radiosensitivities that they observe and compare those values with the available literature.

Answer:

In this work, the intention of the in vitro cell studies was not to obtain cytotoxic effects due to an ablative radiation dose to cancer cells produced by I-131, which is a very well-known issue; nor demonstrate that cells are extremely radiosensitive. Our aim was to evaluate the cytotoxic effect of the I-C19 in radiosensitized cells (synergistic effect between I-C19 and I-131 radiation); that’s why intentionally we prepared 131I-C-19 with a very low specific activity. What we demonstrated is that approximate 0.2 Gy effectively potentiate the cytotoxicity of I-C19 which is the contribution of this research. The inhibition of K-ras4B and its signaling pathways is directly proportional to the radiosensitizing effect on cancer cells [1]. Moreover, radiation absorbed doses at 0.2 Gy are sufficient to induce deterministic effects in human cells, for example, the changes observed in blood cells at 0.2 Gy. 

In order to clarify this point, the following paragraph was added in line 193-198:

“The inhibition of K-ras4B and its signaling pathways is directly proportional to the radiosensitizing effect on cancer cells [1]. Absorbed radiation doses at 0.2 Gy are enough to induce deterministic effects in human cells, for example, the phenotypic changes observed in blood cells. Therefore, our results demonstrated that a dose of approximately 0.2 Gy is sufficient to effectively potentiate I-C19 cytotoxicity (synergistic effect between I-C19 and I-131 radiation).”

[1] Gurtner K, Kryzmien Z, Koi L, Wang M, Benes CH, Hering S, Willers H, Baumann M, Krause M. Radioresistance of KRAS/TP53-mutated lung cancer can be overcome by radiation dose escalation or EGFR tyrosine kinase inhibition in vivo. Int J Cancer. 2020 Jul 15;147(2):472-477. doi: 10.1002/ijc.32598.

Other comments:

Please, notice that section 4.16 “Biodistribution and biokinetic models” and 4.17 “Radiation absorbed dose estimation” are presented twice.

Answer:

Thank you for the observation, the mistake was corrected.

The authors have cleared my issues with the in vivo data. It is somewhat of a pity that the authors did not study tumor volume evolution when treated with the 131I labelled compound. Those data would be really interesting, and I encourage the authors to complete

the study in the future.

Answer:

Thank you for the suggestion, we will do it.

Reviewer 3 Report

See attached file.

Author Response

(The authors gave the same response as above.)
